# Molecular asymmetry of a photosynthetic supercomplex from green sulfur bacteria

Ryan Puskar[1,2], Chloe Du Truong[1,2,7], Kyle Swain[3], Saborni Chowdhury[1,2], Ka-Yi Chan[1,2], Shan Li[4], Kai-Wen Cheng [4], Ting Yu Wang[5], Yu-Ping Poh[2,8], Yuval Mazor [1,2], Haijun Liu [6], Tsui-Fen Chou [4,5], Brent L. Nannenga [2,3] & Po-Lin Chiu [1,2] ✉

The photochemical reaction center (RC) features a dimeric architecture for charge separation across the membrane. In green sulfur bacteria (GSB), the trimeric Fenna-Matthews-Olson (FMO) complex mediates the transfer of light energy from the chlorosome antenna complex to the RC. Here we determine the structure of the photosynthetic supercomplex from the GSB Chlorobaculum tepidum using single-particle cryogenic electron microscopy (cryo-EM) and identify the cytochrome c subunit (PscC), two accessory protein subunits (PscE and PscF), a second FMO trimeric complex, and a linker pigment between FMO and the RC core. The protein subunits that are assembled with the symmetric RC core generate an asymmetric photosynthetic supercomplex. One linker bacteriochlorophyll (BChl) is located in one of the two FMO-PscA interfaces, leading to differential efficiencies of the two energy transfer branches. The two FMO trimeric complexes establish two different binding interfaces with the RC cytoplasmic surface, driven by the associated accessory subunits. This structure of the GSB photosynthetic supercomplex provides mechanistic insight into the light excitation energy transfer routes and a possible evolutionary transition intermediate of the bacterial photosynthetic supercomplex from the primitive homodimeric RC.

Green sulfur bacteria (GSB) contain large light-harvesting antenna structures, known as chlorosomes, which enclose many stacked bacteriochlorophylls (BChls) that collect energy from light excitation for subsequent charge separation in the membrane reaction center (RC)[1,2]. Upon illumination, the excitation energy is sequentially transferred from the chlorosome BChls through a monolayered baseplate[2,3], Fenna-Matthews-Olson (FMO) complexes[4], and ultimately to a membrane-embedded RC, where charge separation occurs. The GSB FMO protein complexes (from *Prosthecochloris aestuarii* and

*Chlorobaculum tepidum*) were the first available high-resolution structures of the photosynthetic antenna[5]. The structure of the FMO complex is a symmetric homotrimer[6–8], and each FMO monomer contains eight BChl *a* pigments with distinct site energies[9,10]. The energy transition between these pigments has been extensively investigated with spectroscopic studies and theoretical calculations[9,11]. The two-dimensional (2D) spectroscopy results showed a long-lived quantum coherence (>300 fs at physiological temperatures[12]), suggesting a superposition of wave-like motions during exciton transfer[13].

[1]School of Molecular Sciences, Arizona State University, Tempe, AZ 85287, USA. [2]Center for Applied Structural Discovery, Biodesign Institute, Arizona State University, Tempe, AZ 85287, USA. [3]School for Engineering of Matter, Transport and Energy, Arizona State University, Tempe, AZ 85287, USA. [4]Division of Biology and Biological Engineering, California Institute of Technology, Pasadena, CA 91125, USA. [5]Proteome Exploration Laboratory, Beckman Institute, California Institute of Technology, Pasadena, CA 91125, USA. [6]Department of Biology, Washington University, St. Louis, MO 63130, USA. [7]Present address: Rampart Bioscience, Monrovia, CA 91016, USA. [8]Present address: Center for Mechanisms of Evolution, Biodesign Institute, Arizona State University, Tempe, AZ 85287, USA. ✉e-mail: plchiu@asu.edu

However, whether the quantum coherence can be maintained in a native protein-crowded environment with thermal fluctuations at physiological temperatures is still unclear[14].

RCs are large pigment-protein complexes that utilize photo-excitation for charge separation[15], and all photosynthetic RC cores are dimeric[16]. The GsbRC is a type I RC that uses iron-sulfur (FeS) clusters as terminal electron acceptors, similar to cyanobacterial and plant photosystem I (PSI) complex[1,17]. Because only one gene encodes two copies of the GsbRC core proteins (PscA subunit)[18], the GsbRC has a homodimeric architecture with a symmetric distribution of pigments, suggesting two identical branches of electron transport chains (ETC) that could be equally utilized[19]. The cyanobacterial PSI RC has two symmetrical pigment branches, but the mutagenesis study and molecular simulations, however, showed that the electron transfer is asymmetric[20–22]. Further experimental investigations are required to understand whether the efficiencies of the two ETC branches in GsbRC are similar to those of the cyanobacterial PSI.

The foundation of the homodimeric platform of the GsbRC is formed by the two PscA membrane subunits, which host a P840 electron donor, a primary acceptor Acc, a possible secondary acceptor $A_0$, an FeS center X ($F_X$), and multiple BChls[1,23–25]. Two membrane-associated PscC subunits (cytochrome $c_z$ or cyt $c_{551}$) contain the cytochrome $c$ domain, which mediates electron transfer from the menaquinol/cytochrome $c$ oxidoreductase to the P840[26–29]. On the cytoplasmic surface, PscB binds two FeS clusters ($F_A$ and $F_B$), serving as terminal electron acceptors[30]. PscD is involved in ferredoxin docking and is homologous to the PsaD in PSI from cyanobacteria and plants (-9.8% sequence identity to the PsaD of *Synechocystis* sp. PCC 6803)[31]. A previous cryo-EM structure of the GsbRC showed the subunit organization of the PscA, PscB, PscD, and one FMO trimer in the photosynthetic supercomplex[32].

Although the previous RC-FMO complex structure reveals how one FMO trimer associates with the RC[32], some questions remain unanswered. The PscC subunit was missing in the structure, hindering the complete understanding of the ETC electron donor in the GSB photosynthetic supercomplex[32]. In addition, previous biochemical characterization has shown more than one FMO complex attached to the RC[1,33]. Although it was suggested that the space constraints of the RC cytoplasmic surface would reasonably allow for an additional FMO trimer to bind[32], it is still unclear how multiple FMO complexes could organize on the RC cytoplasmic surface. Moreover, it is worth investigating the functional role of the FMO complex if the RC core can adopt more than one FMO complex.

Here we characterize the photosynthetic supercomplex purified from *C. tepidum* and determine its high-resolution structure using single-particle cryo-EM. In contrast to the previous approach for protein purification[32], we utilized a mild detergent for membrane protein extraction to reduce disruptions to the interactions between proteins and lipids. This approach preserved a more complete photosynthetic complex, which could be partly due to the bound membrane lipids maintaining the integrity of the protein complex[34]. The purified sample contains complexes with different stoichiometries of RC and FMO proteins, and the cryo-EM reconstructions suggest that the RC can host at most two FMO trimer complexes on its cytoplasmic surface. Our RC-FMO$_2$ reconstruction also reveals the structure of the associated PscC subunits, herein-identified accessory subunits, and pigment molecules. These accessory subunits bind to only one side of the dimeric RC core, creating asymmetric binding interfaces for two FMO trimers. A linker pigment is identified at the interface between the FMO and PscA subunit, which is only found in one of the FMO-PscA axes. Combining all findings from our structure, the subunit association builds an asymmetric photosynthetic supercomplex on the symmetric RC core, which could lead to two pathways with differential efficiencies for energy transfer.

## Results and discussion

### Various stoichiometries of FMO and RC

Detergents used for membrane protein solubilization and extraction have been reported to affect the stability of the photosynthetic supercomplex of *C. tepidum*[1,26,35–39]. The non-ionic maltoside-based detergent, *n*-dodecyl-β-D-maltoside (DDM), has previously been shown to preserve a majority of the photosynthetic complex during protein extraction[40,41]. Here, the resulting DDM-solubilized and purified samples were characterized by size-exclusion chromatography and polyacrylamide gel electrophoresis (SDS-PAGE) (Supplementary Fig. 1a, b). Mass spectrometry (MS) analysis of the bands excised from blue native-gel electrophoresis (BNGE) indicated individual subunits that compose the photosynthetic complex, including two herein-identified protein subunits (Supplementary Tables 1, 2; Accession numbers: Q8KDI3 and Q8KG87). Negative-stain EM showed that the protein complexes were heterogeneous but had a shape that is consistent with a dimeric complex (Supplementary Fig. 1c). We subsequently performed single-particle cryo-EM analysis, sorted the particles into different categories, and characterized their three-dimensional (3D) structures (Supplementary Figs. 2, 3). The two-dimensional (2D) particle images were reconstructed into cryo-EM densities with different assembly states: RC-FMO$_2$ (one RC core with two FMO trimeric complexes), RC-FMO (one RC core with one FMO trimeric complex), and RC alone, which showed the varied stoichiometries possible for the RC and FMO complex (Supplementary Fig. 3). These results corroborate previous observations on the varied stoichiometries of the RC and FMO protein complexes in purified samples following crosslinking, LC-MS/MS, and scanning transmission electron microscopy (STEM)[37,40]. Although various stoichiometries of the RC and FMO have been reported[33], we did not identify any reconstructions that consist of more than two FMO trimers bound to the RC in our dataset. This indicates that the two FMO trimers bound onto the RC seem to be the maximum[1,32], which is in agreement with the limited space available on the RC cytoplasmic surface. The varied stoichiometries seen here suggest a dynamic or differential interaction between the RC and FMO protein complexes in their native conditions. Also, it is known that the FMO complexes are sandwiched between the chlorosome baseplate and membrane RC in the cell, therefore it is possible that in the absence of the baseplate, the FMO proteins are not stably bound to the RC, leading to the various stoichiometries seen for the RC-FMO complexes.

### Cryo-EM reconstruction of the GsbRC-FMO supercomplex

The 3D cryo-EM densities of the RC-FMO$_2$ and RC-FMO$_1$ complexes were reconstructed at 3.08 Å and 3.49 Å resolution, respectively (Supplementary Fig. 3). Further signal subtraction and local refinement improved the two local regions of the RC-FMO$_2$ density up to 2.92 Å (FMO1, PscA1, and PscB) and 3.06 Å (FMO2 and PscA2) (Fig. 1; Supplementary Figs. 3, 4). Individual subunits in the photosynthetic complex could be identified, and their atomic coordinates were built within the cryo-EM densities (Supplementary Fig. 5). When our RC-FMO$_1$ structure was superimposed with the previous cryo-EM structure, no significant structural differences could be identified (Supplementary Fig. 6a, b)[32]. However, our RC-FMO$_2$ structure reveals the association of two FMO trimers, two PscC subunits, two accessory membrane subunits, and additional pigment molecules (Fig. 1). Also, when the RC-FMO$_2$ structure is compared to the RC-FMO$_1$ or the previous cryo-EM structure[32], a loop of PscD (Q83-P108) exhibits a slight conformational change in the presence of the second FMO trimer. This structural change could accommodate and stabilize the binding of the second FMO trimer (Supplementary Fig. 6c-f).

### Visualization of PscC and PscE

The structures of the RC dimeric cores in the RC-FMO$_2$ structure do not exhibit noticeable variations among the three structures

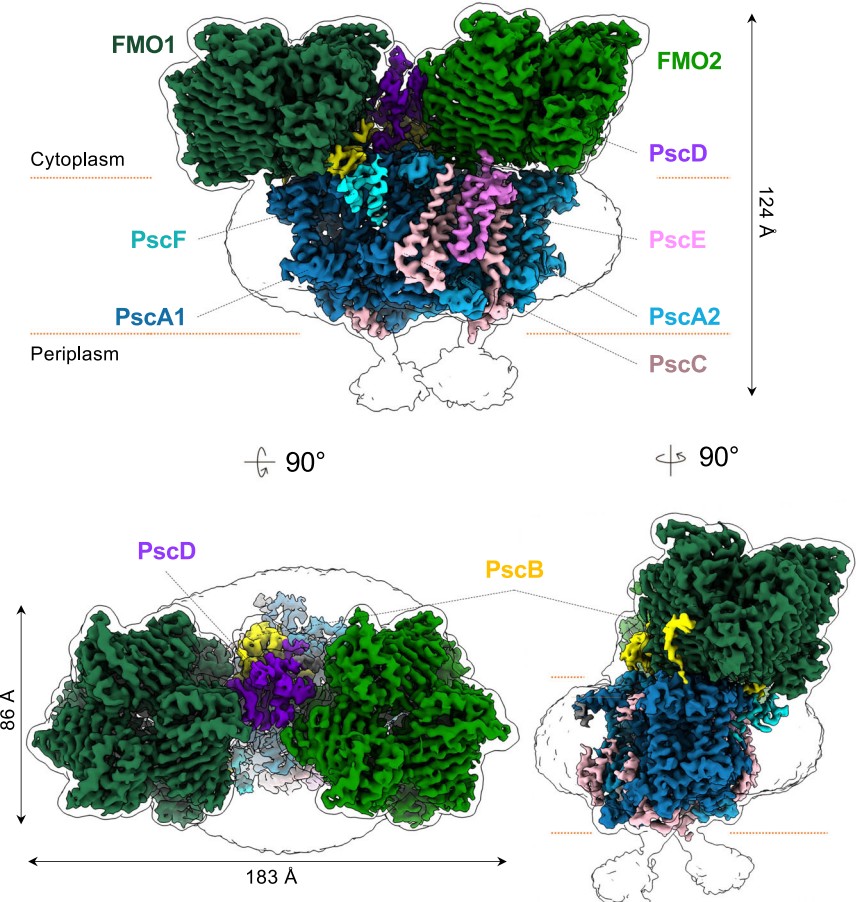

**Fig. 1 | Cryo-EM reconstruction of the photosynthetic supercomplex from *Chlorobaculum tepidum*.** Three-dimensional (3D) cryo-EM density map of the RC-FMO₂ assembly. Color codes: FMO1−dark green; FMO2−forest green; PscA1−blue; PscA2−light blue; PscB−yellow; PscC−light pink; PscD−purple; PscE−magenta (UniProt accession code: Q8KDI3); and PscF−cyan (UniProt accession code: Q8KG87). A 6 Å⁻¹-filtered surface envelope (1.0σ) is overlaid over the density of the protein supercomplex (3.6σ). Horizontal dashed lines (orange) indicate membrane boundaries.

(Supplementary Fig. 6a–c), and we were able to identify the cytochrome *c* subunit, PscC, in our cryo-EM density maps. PscC, which serves as an electron donor to reduce P840 in the RC, features three N-terminal transmembrane helices with a C-terminal heme-binding domain (Supplementary Fig. 7a)[40,42–44]. In both RC-FMO₂ and RC-FMO₁ density maps, the two PscC subunits bind peripherally on both sides of the PscA dimer symmetrically (Fig. 1; Supplementary Fig. 7b). The PscC density that associates with PscA1 has a low signal content, especially the αC1 helix, and the two PscC subunits possibly have different binding occupancies to the PscA subunit. The PscC that binds to PscA2 has a higher signal level (>4.0σ) and interacts with a herein discovered helix-turn-helix subunit (59 residues modeled; UniProt accession number: Q8KDI3), which attaches to the FMO2 surface (Fig. 1; Supplementary Tables 1, 2). This membrane subunit contains a high proportion of charged residues (aspartate and glutamate: 17%; lysine and arginine: 22%) (Supplementary Table 3). Following the nomenclature for the GsbRC[15], we hereafter refer to this protein subunit as PscE. Unlike PscC, only one copy of PscE was identified in the complex. Because the PscE is associated with PscC and FMO2 but was lost in the RC-FMO₁ or the previous structure[32], it likely plays an important role in stabilizing the association of PscC and FMO2.

The cryo-EM density of the C-terminal heme-binding domain was resolved at a low resolution, and only visible at lower contour levels (1.0σ) (Supplementary Fig. 7b). These two low-resolution envelopes on the RC periplasmic surface can be identified in both our RC-FMO₂ and RC-FMO₁ densities (Supplementary Fig. 7b). The atomic coordinate of the PscC heme-binding domain (PDB code: 3A9F) can be fit into these

densities, showing the position of the domain attached to the PscA periplasmic surface close to the P840 center (Supplementary Fig. 7c)[29]. The low-scattering content of the C-terminal heme-binding domain implies its mobile nature and its relatively weak affinity to the PscA surface[27,45].

### A linker pigment and PscF at the FMO-RC interface

A herein-identified accessory subunit, composed of four membrane helices that are fully embedded in the bilayer, interacts with FMO1, PscA1, and the C-terminus of the PscB (Figs. 1, 2a). Like PscE, a single copy of PscF was found attached to the RC. It was also detected by the mass spectrometry (MS) proteomic analysis (UniProt accession number: Q8KG87; annotated as Ric1), which has the best-fit model to the cryo-EM density (Supplementary Fig. 5 and Tables 1, 2). The STRING database for protein-protein interactions shows that this protein subunit interacts with PscB and cytochrome *c* proteins (PscC)[46], consistent with the spatial organization shown in the cryo-EM reconstruction.

A BChl pigment (BChl-A816) and a monogalactosyl diglyceride (MGDG) lipid are located between PscF and PscA1 (Fig. 2a). PscF and BChl-A816 were not found in the RC-FMO₁ or the previous cryo-EM structure[32], possibly due to their weak association with the complex. The MGDG headgroup interacts with FMO1 via hydrogen bonds (H13 and K35), and its acyl chains interact with BChl-A816 through hydrophobic interactions (Fig. 2a). The MGDG lipids can be found in the RC-FMO₁ structure, but not the BChl-A816 or PscF. Thus, the interactions with PscB, PscF, and the MGDG lipid assist in positioning the BChl-A816 at the interface of the FMO1 trimer and PscA1 subunit.

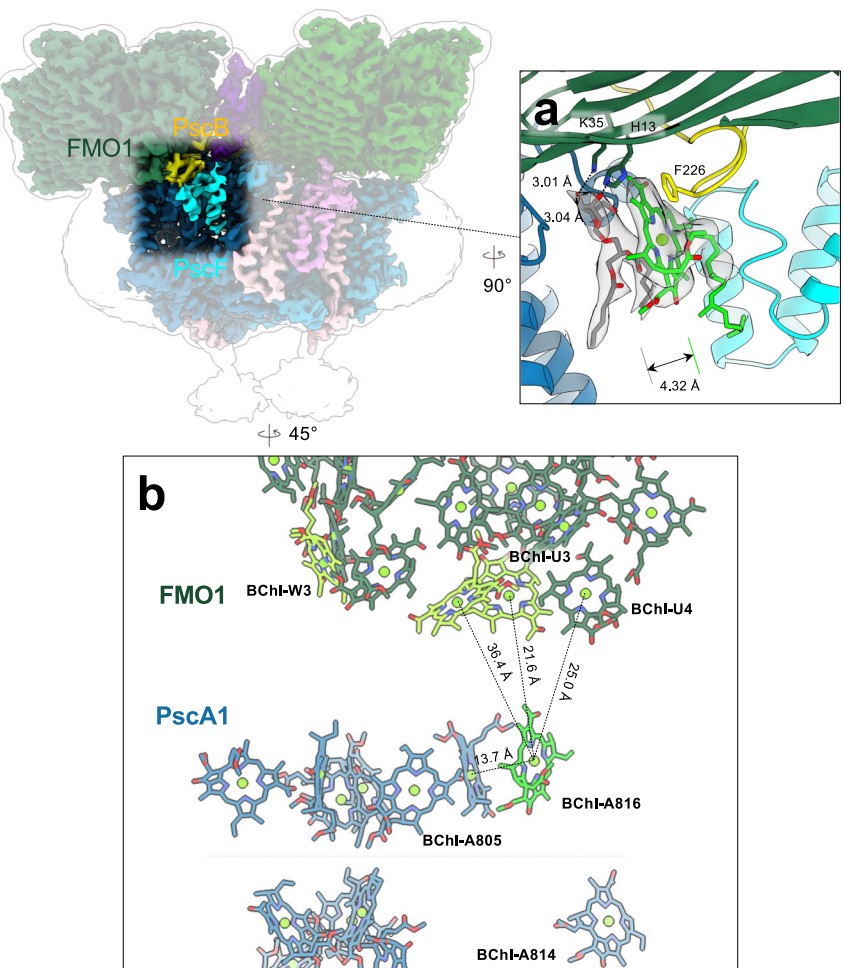

**Fig. 2 | Potential linker pigment at the interface between the FMO complex and the RC core. a** Potential linker bacteriochlorophyll (BChl) at the FMO and RC interface. A BChl *a* (BChl-A816; light green) and an MGDG (monogalactosyl diglyceride) lipid (gray) sandwiched between PscF (cyan) and PscA1 (blue) subunit. Gray surfaces are cryo-EM densities of the BChl and MGDG molecules (3.5σ). **b** Spatial arrangements of the BChl pigments within the FMO1 and PscA1 subunit. The BChl-A816-BChl-U3 has the shortest Mg–Mg distance (21.6 Å) among the BChl pairs within the FMO1·PscA1 subunit. Orange dashed line separates the cytoplasmic and periplasmic layers of BChl clusters.

The BChl site energies define the direction of energy flow[9], and the distance between BChls is one of the critical factors that determine the efficiency of the Förster energy transfer[47] or exciton coupling[48]. BChl-A816 is located near the site-3 BChl *a* in FMO1 (BChl-U3) (Mg–Mg distance: 21.6 Å), which has the lowest site energy among all the other seven BChls in the FMO protein (Fig. 2b)[9]. In the previously reported structure that lacks this linker BChl (PDB code: 6M32), the shortest Mg–Mg distance between the BChls in FMO1 (chain F, site 3) and PscA (A810) is 29.0 Å[32]. At the FMO2-PscA2 interface, the shortest Mg–Mg distance between BChls is 28.1 Å (BChl-Y3-BChl-a808) (Supplementary Fig. 8). In addition, the distance between the BChl-A816 and its closest neighbor BChl in the PscA1 (BChl-A805) is 13.7 Å. Therefore, this identified BChl is very likely to serve as a linker bacteriochlorophyll to mediate exciton transfer from FMO1 to the RC. This linker BChl is not found at the FMO2 and PscA2 interface, and the spaces between PscA2, PscC, PscE, and FMO2 are filled with lipid molecules, leaving no space for additional BChls.

Calculating the transfer rates between FMO1 and RC using Förster theory further supported the significance of BChl-A816 in the transfer process. In addition to its close proximity, the orientation between the BChl-A816 and the terminal emitters of the FMO (subunit U) is highly favorable for energy transfer, leading to the fastest transfer rate in the entire FMO-RC interface (Supplementary Fig. 9 and Supplementary Data 1, 2). A simple Förster treatment using uniform site energies yields

the FMO-to-RC transfer rate of ~0.13 ps$^{-1}$, significantly faster (~50×) than in vitro measurements, but in agreement with some recent in vivo estimations[49,50]. The discrepancy can be attributed to the lability of some pigments, such as BChl-A816, during isolation or to some limitations of the Förster treatment resulting in an overestimation of the transfer rate. Regardless of the absolute value of the transfer rate, the identification of BChl-A816 as a major contributor to energy transfer in this system should hold. It is also quite clear that FMO2 is not as well connected to the RC as FMO1, this is due to both the absence of BChl-A816 and slower overall transfer rates caused by the larger distance separating FMO2 and the RC (Supplementary Fig. 9 and Supplementary Data 1, 2).

**Asymmetric binding of the two FMO complexes**
In the RC-FMO$_2$ supercomplex, the orientation of the two FMO trimers does not follow the symmetry of the PscA dimer, and each associate with the RC through a unique binding interface (Fig. 3). The two FMO trimers are 97 Å away without direct contact and are bridged by the PscB and PscD subunits (Fig. 3). The two $C_3$ symmetrical axes of the FMO trimers tilt 12.6° (FMO1) and 1.3° (FMO2) against the normal axis of the membrane plane (Fig. 3; 15° tilt for the FMO1 in the previous cryo-EM structure[32]). The residues that contact FMO2 are more distant than those contacting FMO1, and the effective areas that contact FMO2 are much smaller than those for FMO1, resulting in a larger solvent-

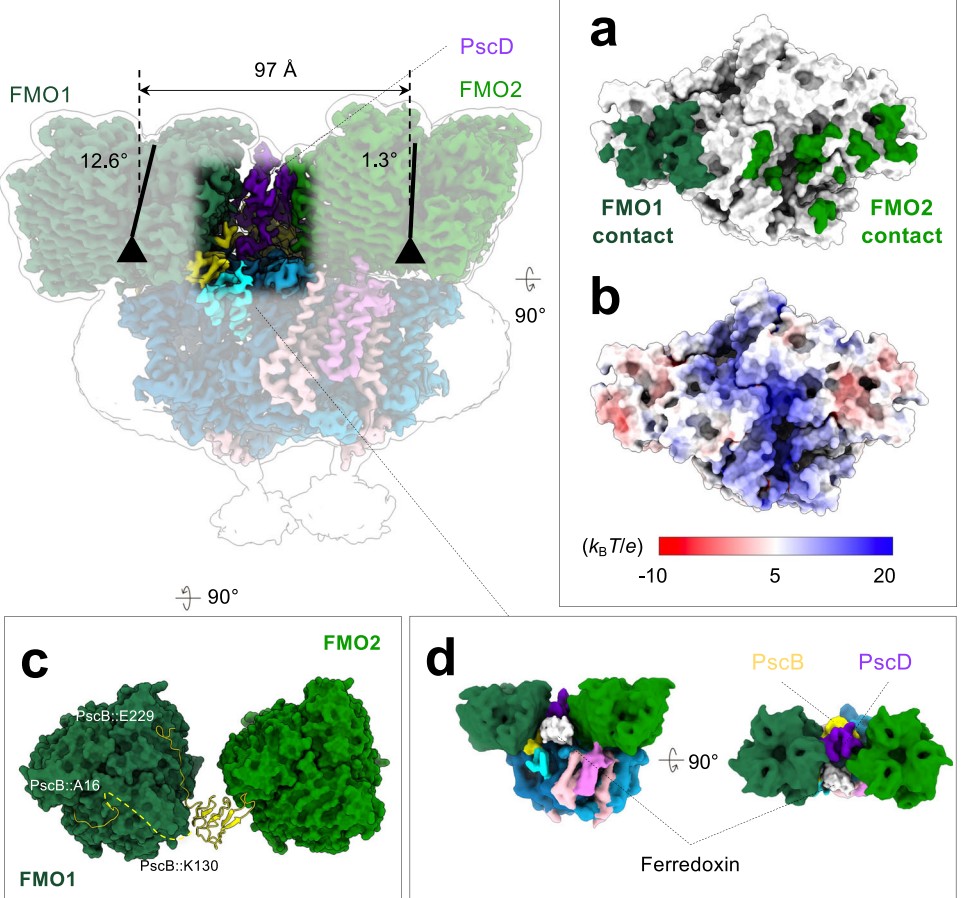

**Fig. 3 | Asymmetric binding of the two FMO complexes on the RC cytoplasmic surface.** The gravity centers of the two FMO trimers are separated by 97 Å. Three-fold axes (black triangle with an axis) of the FMO trimers are tilted to the normal of the membrane plane at 12.6° (FMO1) and 1.3° (FMO2), respectively. **a** Contact of the two FMO trimers on the RC cytoplasmic surface. Surface representation includes the cytoplasmic side of PscA, PscC, PscE, and PscF. Residues that contact the FMO complexes within 5 Å are colored in dark green (FMO1) and forest green (FMO2).

**b** Electrostatic potential map of the RC cytoplasmic surface. Values are presented as the energy per unit charge ($k_BT/e$). Color bar indicates blue and red as positive and negative values, respectively. **c** PscB (yellow) has a larger contact with the FMO1 (dark green) than FMO2 (forest green). Yellow dashed curve represents the unmodelled loop from residue 17–129 of PscB. **d** Schematics of the potential fer-redoxin docking site. White surface represents the ferredoxin enzyme (UniProt accession number: Q8KCZ6).

excluded surface for FMO1 with the membrane RC (624.0 Å²) than that for FMO2 (247.1 Å²) (Fig. 3a). The electrostatic potential map shows that the FMO binding sites on the RC surface are more negatively charged compared to the overall surface of the center (Fig. 3b). Note that the FMO1 is the one seen in the RC-FMO₁ structure as well as in the previous structure[32]. Different binding affinities between the two FMO trimers have been reported or discussed previously[32,37,40,51]. Combined with the structural observations, FMO1 seems to be more stably bound to the RC cytoplasmic surface than FMO2.

The interactions between PscB and the two FMO trimers are also different. PscB has a larger contact area with FMO1 than FMO2 (Fig. 3c). The N- and C-terminal loops of PscB bind the FMO1 surface and wrap around the FMO1 trimer, likely stabilizing the FMO1 association in the complex assembly. However, PscB has a smaller contact area with the FMO2 (Fig. 3c), further differentiating the binding affinities of the two FMO complexes with the photosynthetic supercomplex.

The linker pigment was identified along the FMO1-PscA1 axis, but not the FMO2-PscA2 axis (Fig. 2a), and this could result in a higher energy transfer efficiency in the FMO1-PscA1 pathway than the FMO2-PscA2. Because the orientation of the FMO trimer on the RC determines the distances between FMO-BChls and RC-BChls[52], the tilt angle of the symmetrical axis of the FMO1 may need to be maintained for optimal energy transfer efficiency (Fig. 3). Thus, the FMO2 trimer may play a role in supporting the intermembrane

space to maintain the orientation of FMO1 relative to the membrane and RC core.

In the cell, the intermembrane space between the chlorosome baseplate and plasma membrane is confined by the sandwiched FMO complexes. In the RC-FMO₂ supercomplex, the binding of PscB and PscD with the two FMO trimers leaves one side in the cytoplasmic space available for ferredoxin docking (Fig. 3d)[1], and the size of this open space is sufficient for the entry of one ferredoxin molecule (Fig. 3d; Supplementary Fig. 10a–d). The conserved K30 side chain of the PscD subunit also points toward this potential ferredoxin docking site and is possibly involved in the binding of the ferredoxin (Supplementary Fig. 10d)[31].

**Asymmetric pigment distribution in the supercomplex**
In our RC-FMO₂ structure, a total of 25 BChl *a* (13 bound to PscA1 and 12 bound to PscA2), two BChl *a*', four Chl *a*, three chlorobactenes (F26), and three chlorobactene glucoside laurate (F39) pigment molecules in the RC core were modeled (Fig. 4a). Forty-eight BChl *a* were modeled in the two FMO trimers (Fig. 4a). Compared to the pigment numbers of the anaerobic heliobacterial reaction center *Hb*RC (*Heliomicrobium modesticaldum*) (60) or plant PSI (87), the RC from *C. tepidum* has much fewer pigments within its RC[53–55]. Compared to the RC-FMO₁ structure presented here or the previous cryo-EM structure[32], the RC-FMO₂ structure has revealed additional pigments: one BChl *a* (located

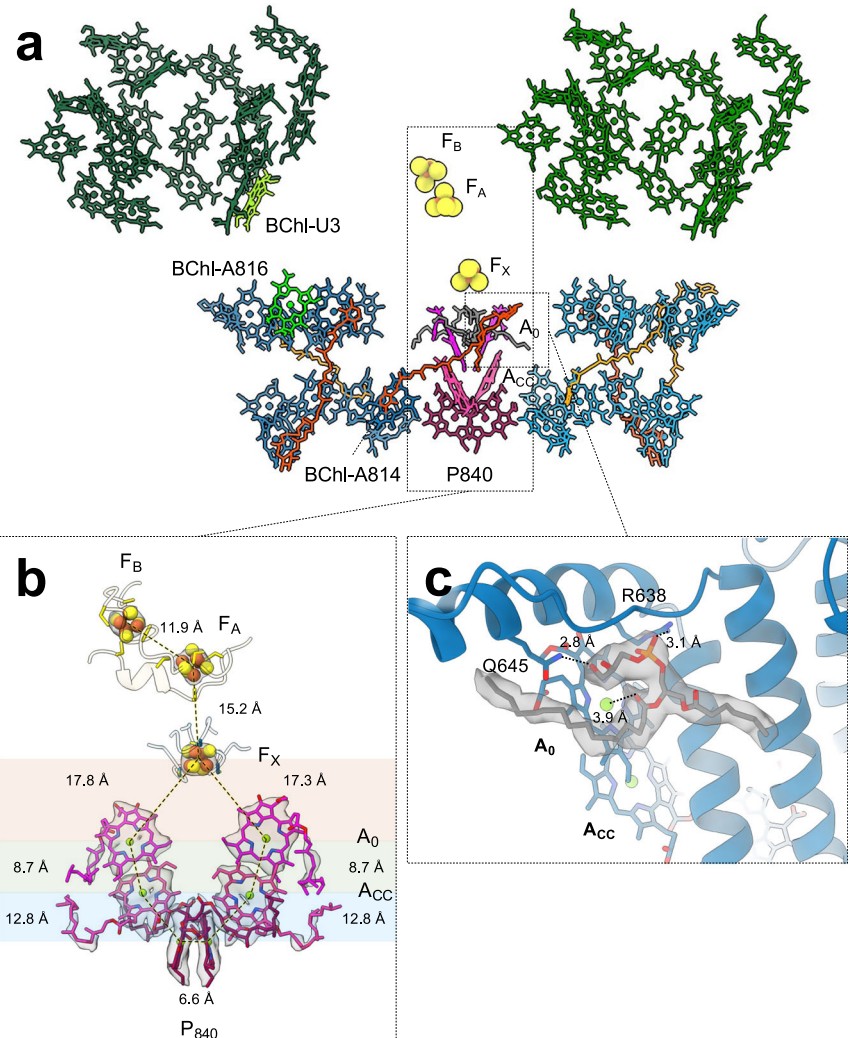

**Fig. 4 | Pigment distribution in the RC-FMO₂ photosynthetic supercomplex.** **a** BChl *a* molecules are colored in dark green (FMO1), forest green (FMO2), blue (PscA1), and light blue (PscA2). BChl-U3 and BChl-A816 are colored in green yellow and light green, respectively. Pigments for P840, $A_0$, and $A_{CC}$ centers are colored in purple, dark pink, and magenta, respectively. F39 and F26 carotenoids are colored in orange red and orange sticks, respectively. The central F39 carotenoid at the PscA dimer interface has a tilt angle of 54° to the normal of the membrane plane. Two phosphatidyl glycerol (PG) lipids near $A_0$ are shown in gray sticks. Iron-sulfur (FeS) clusters are presented in yellow and orange spheres. **b** Electron transport chain (ETC) in the RC-FMO₂ assembly. Cartoons are the protein residues that interact with the FeS clusters. Cryo-EM densities of the pigments are shown in gray surfaces (4.0σ). **c** Interaction of the embedded PG lipid with the PscA1 and $A_0$ chlorophyll. The PG headgroup is stabilized via hydrogen bonding with R638 and Q645 side chains of the PscA1 subunit. The carbonyl group of the lipid glycerol backbone interacts with $A_0$ chlorophyll. Gray surface shows the cryo-EM density of the PG lipid (1.8σ).

at the interface between FMO1 and PscA1) (Fig. 2a), one F39 (located near the PscA dimer interface) (Fig. 4a; Supplementary Fig. 11a), and one F26 (located in the PscA2) (Fig. 4a; Supplementary Fig. 11b). As seen previously for the *Hb*RC and the GsbRC[32], we did not identify any quinone molecules in the GsbRC structures.

The FeS clusters and P840 dimer that drive the ETC in RC-FMO₂ are arranged similarly to the RC-FMO₁ and the previously reported structure (Fig. 4b)[32]. The inter-cofactor distances for the $A_0$-$F_X$ (17.8 Å and 17.3 Å) and Acc-$A_0$ (8.7 Å and 8.7 Å) pairs are slightly shorter than in the previous structure ($A_0$-$F_X$: 18.2 Å and 18.1 Å; Acc-$A_0$: 10.0 Å and 10.0 Å) (Fig. 4b)[32]. In contrast to the structure of *Hb*RC[53], we did not find a water molecule as an axial ligand to the Mg(II) of $A_0$; instead, we found two phosphatidyl glycerol (PG) lipids interacting with the $A_0$ pair within the PscA dimer (Fig. 4c). These PG lipids also fit the unmodelled densities between $A_0$ and $F_X$ in the previous GsbRC-FMO₁ cryo-EM reconstruction[32] (Supplementary Fig. 12a). In the *Hb*RC electron density, two unmodelled densities were found in similar positions and were initially suggested to be isoprenyl phosphates[53] and later

modeled as PG lipids[45]. However, our PG coordinates do not fit into the unmodelled densities in the *Hb*RC structure (Supplementary Fig. 12b). They may present a slightly different conformation but possibly the same function as the PG lipids in the GsbRC-FMO₂ at this site. In our GsbRC-FMO₂ structure, the model coordinates of PG fit our densities (local cross-correlation coefficient 0.803), and the ester carbonyl oxygen of the PG lipid serves as the axial ligand to $A_0$ (3.9 Å) (Fig. 4c). The PG headgroup locates more closely to the membrane center than those of other lipids and is stabilized by hydrogen bonds with the side chains of R638 and Q645 (Fig. 4c). The lipid acyl chains interact with neighbor lipids, protein hydrophobic residues, and the $A_0$ porphyrin ring (Fig. 4c). It is uncertain about the function of this PG lipid, but its interaction with $A_0$ may play a role in modulating the ETC efficiency.

The carotenoids in the photosynthetic complex are suggested to serve as photoprotection under high-light conditions[56–58]. All identified carotenoid molecules are located in the PscA dimer as previously observed[59–61]. One herein-identified F39 carotenoid molecule is found near the interface of the two PscA dimers. The distance between the

F39 trimethylbenzene and the BChl-A814 porphyrin rings is 3.4 Å (Supplementary Fig. 11c). The F39 chlorobactene chain and glycosyl tail are sandwiched between the PscA1 and PscC subunits and extensively interact with lipid acyl chains (Supplementary Fig. 11c). These interactions provide a highly hydrophobic environment for stabilizing F39 binding[62]. On the opposite side of the PscA dimer, partly because of the lower occupancy of PscC, F39 does not bind strongly to PscA2, and its density is not seen in our cryo-EM map. Also, because our cells were grown in a high-light condition (~350 μmol photons/m$^2$/sec), the additional carotenoids discovered in the RC-FMO$_2$ complex could assist in dissipating excessive light energy to adapt to the light stress.

### Membrane lipids stabilize the supercomplex

The previous cryo-EM structure of the RC-FMO complex did not identify bound PscC subunits[32], and it is possible that the detergent used in the previous study interrupted the lipid-protein interactions and destabilized PscC binding. Ten PG and six MGDG could be modeled in our RC-FMO$_2$ cryo-EM density map (Supplementary Fig. 13). Furthermore, lipids were found to be located between protein subunits, suggesting that lipids may play a critical role in maintaining the integrity of the photosynthetic supercomplex (Supplementary Fig. 13). To stabilize PscC in the photosynthetic complex, two of the transmembrane helices (αC1 and αC2) bind to the PscA dimer interface and one (αC3) extensively interacts with PscA transmembrane helices (Supplementary Figs. 7b, 13). These three transmembrane helices extensively interact with membrane lipids, especially in the cytoplasmic leaflet (Supplementary Fig. 13). In the space between PscC, PscA, and PscE subunit, we were able to model one lipid in the periplasmic leaflet and seven lipids in the cytoplasmic leaflet, all of which are stabilized by hydrophobic interactions.

### Molecular asymmetry biases the energy transfer

It is an open question whether the two branches of the energy transfer pathway within a bacterial photosynthetic complex have the same efficiency. Our structure of the photosynthetic supercomplex with the two FMO trimers of *C. tepidum* (RC-FMO$_2$) reveals the molecular asymmetry in subunit association and pigment distribution. Compositions of the pigments and spatial arrangements of the antenna proteins can change the energy transfer pathway and modulate the transfer efficiencies[63]. Although the GsbRC core has a homodimeric architecture, most of the subunits that are associated with the dimeric RC do not follow the two-fold symmetry of the RC core, which is likely to differentiate the efficiencies between the two branches. Nearly no loss was reported for energy transfer from the chlorosome to FMO proteins, but the efficiency is reduced to 75% or less for transfer from FMO to the RC[49,64–66]. This is consistent with a hypothesis in which one of the energy transfer pathways from FMO to RC has higher efficiency, and the other has lower efficiency, yielding a weighted average of ~75% for overall transfer efficiency.

The previous structural and phylogenetic study showed that the RCs have a highly conserved structural core even though their sequences are divergent[67]. This suggests that the selective constraints of maintaining the homodimeric RC are robust[67]. In most cases, the symmetric arrangement of a protein could offer the advantage of cooperativity[68]. However, the structure of the RC-FMO$_2$ supercomplex shows that the membrane subunit association can break the symmetry of the overall supercomplex, resulting in unequal pigment distributions and different efficiencies for the two energy transfer pathways. These small protein subunits are most likely expressed to stabilize the structure and modulate the function of the protein complex, which could be important to the understanding of how the biomolecular complex evolves to adapt to environmental change.

Why the asymmetric arrangement of protein subunits in the GsbRC-FMO$_2$ supercomplex is established on a homodimeric RC core? Because our protein complex was directly extracted from the native

membranes, the probability of obtaining a complex randomly formed by protein subunits is likely low. However, the protein isolation procedure may affect the stability of the protein assembly. Also, many projections of the protein complex imaged by cryo-EM could be used to calculate a high-resolution 3D reconstruction, which shows consistent observations across these molecular images. 3D image classification of our cryo-EM data results in the RC-FMO1-FMO2 (RC-FMO$_2$) and RC-FMO1 (RC-FMO$_1$) densities, but not RC-FMO2 (Supplementary Fig. 3). To test whether the RC-FMO2 could be possibly formed in the sample, we performed the supervised classification with the three densities, and the result corroborates that the RC-FMO2 less likely presents in the population (Supplementary Fig. 14), implying that the FMO1 binding is required for FMO2 binding. Also, due to the different binding interfaces of the PscB with FMO1 and FMO2, it is unlikely that the two FMO trimers symmetrically bind to the RC. We thus propose a model of the supercomplex assembling process (Supplementary Fig. 15). The small membrane subunits, PscE and PscF, are transiently accessible to both sides of the homodimeric RC core. When a PscF binds to one side of the RC core, it may assist in stabilizing the FMO1 attachment, which subsequently determines the binding locations for PscE and FMO2 on the RC core (Supplementary Fig. 15). The densities for PscE or PscF on the other side of the RC core were not present in our cryo-EM density map. The reason could be that on the other side of the RC core, the assembly of these small subunits is not stable without interacting with a third FMO trimer, which is unlikely to be hosted in the supercomplex due to the limited RC cytoplasmic surface. Therefore, the overall asymmetric feature of the GsbRC-FMO$_2$ supercomplex is generated sequentially, but not randomly, through interactions between these proteins.

We propose a possible pathway for the energy transfer and ETC of the RC-FMO supercomplex in *C. tepidum* (Fig. 5). The light-excited energy is collected by the stacks of bacteriochlorophylls in the chlorosome and transferred through the baseplate to the FMO proteins. The energy flows within the FMOs from the bacteriochlorophyll with the highest site energy (site 1) to the one with the lowest site energy (site 3)[9]. The $C_3$ symmetric axis of the FMO1 trimer has a larger tilting angle to the normal of the membrane than that of the FMO2, leading to a distance of ~20 Å between the FMO BChl of site 3 and the linker BChl (BChl-U3 and BChl-A816) (Fig. 2a). On the FMO2 side, because the BChls in FMO2 and PscA2 have larger distances than those in FMO1 and PscA1, a lower probability for exciton transfer may occur along the FMO2-PscA2 axis. Subsequently, the energy transfer on the FMO1-PscA1 axis has higher efficiency, ultimately arriving at the P840 center for charge separation. Although the function of the second FMO trimer is not certain, it may provide structural support to maintain the tilt of the FMO1 symmetrical axis relative to the membrane plane. The charge transfer will begin from the P840 via Acc and A$_0$ to the FeS clusters. The electron will then be transferred to ferredoxin, docking in the location enclosed by the two FMO trimers, PscB, and PscD, for further downstream energy production.

Our cryo-EM structure of the GsbRC-FMO$_2$ molecular supercomplex casts light on the energy transfer process in the bacterial photosynthetic machinery and highlights the asymmetric nature of the subunit association and pigment distribution. Single-particle cryo-EM allowed us to probe these asymmetric features in native-like conditions with an expanded set of antenna subunits, providing mechanistic insights into a possible pathway for exciton flow and energy transfer in the primitive photosynthetic system.

## Methods

### Culturing Chlorobaculum tepidum

Frozen green sulfur bacteria (GSB), *Chlorobaculum tepidum* (*C. tepidum* TLS, DSM 12025), were a gift from Dr. Haijun Liu (Washington University, St Louis, MO). Cells were grown anaerobically in 1 L glass bottles at 40 °C under a white-light illumination of 350 μmol photons/m$^2$·sec

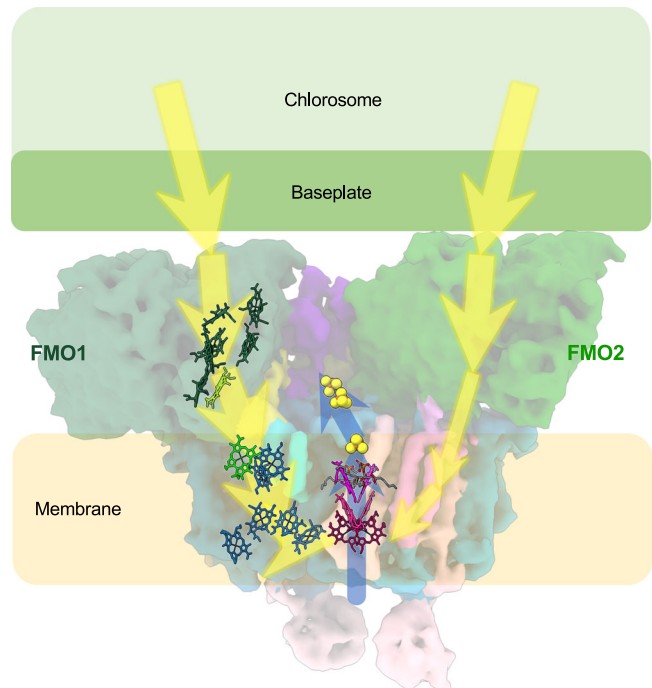

**Fig. 5 | Proposed energy transfer and ETC pathways in the GsbRC-FMO₂ photosynthetic supercomplex.** Model for the energy transfer in the GsbRC-FMO₂ supercomplex. Yellow arrows are possible light excitation energy transfer pathways. Blue arrows indicate the direction of the electron transport along the chlorophylls and iron-sulfur clusters.

for 2–4 days. Cells were harvested at $OD_{750}$ of 2.8 cm⁻¹ and were pelleted using centrifugation at 5000 × $g$ for 7 min. Cell pellets were stored at −80 °C if not used immediately.

## Membrane and protein sample preparation

Cells were resuspended in a buffer of 20 mM Tris-Cl (pH 8.0) and kept at 4 °C throughout all purification steps. A protease inhibitor cocktail tablet (cOmplete, Millipore Sigma, Burlington, MA) was added to the resuspended cells, which were subjected to cell lysis using a sonicator. Cellular debris was removed using centrifugation at 30,000 × $g$ at 4 °C for 15 min, and the supernatant was further centrifuged at 105,000 × $g$ at 4 °C for 1 h. The insoluble pellet was solubilized at 4 °C for 2 h in a buffer of 20 mM Tris-Cl (pH 8.0) and 34 mM $n$-dodecyl-β-D-maltoside (DDM) (Avanti Polar Lipids, Alabaster, AL). The supernatant, after detergent solubilization, was then collected using centrifugation at 105,000 × $g$ at 4 °C for 1 h and was immediately loaded onto a HiTrap Q HP column (Cytiva, Marlborough, MA) in a buffer of 20 mM Tris-Cl (pH 8.0) and 0.85 mM DDM. The eluted fraction at 200 mM NaCl was collected, followed by size-exclusion chromatography using a Superose 6 Increase 10/300 GL column (Cytiva, Marlborough, MA) in a buffer of 20 mM Tris-Cl (pH 8.0), 200 mM NaCl, and 0.17 mM DDM. Absorbance in the wavelength range between 280 and 600 nm was used to detect the target fraction. The peak fraction of the protein complex was collected for subsequent cryo-EM structural studies. Purified protein samples were prepared using the NativePAGE Sample Prep kit (Thermo Fisher Scientific, Waltham, MA) for blue native-gel electrophoresis (BNGE) at a voltage of 150 V and temperature of 4 °C. The gel band was used for further mass spectrometry (MS) analysis.

## LC-MS/MS analysis

Gel bands of the purified sample from native-gel electrophoresis or SDS-PAGE were excised and digested using a trypsin-profile IGD kit (Sigma–Aldrich, St. Louis, MO) according to the manufacturer's

protocol. After desalting and drying, peptides were suspended in a buffer containing 0.2% formic acid and 2% acetonitrile for further LC-MS/MS analysis. LC-MS/MS analyses of the digested peptides from native and SDS-PAGE gels were performed on an EASY-nLC 1200 (Thermo Fisher Scientific, Waltham, MA) coupled to a Q Exactive HF hybrid quadrupole-Orbitrap mass spectrometer (Thermo Fisher Scientific, Waltham, MA) and a Vanquish Neo UHPLC system (Thermo Fisher Scientific, Waltham, MA) with an Orbitrap Eclipse Tribrid mass spectrometer (Thermo Fisher Scientific, Waltham, MA), respectively. Peptides were separated on an Aurora UHPLC column (25 cm × 75 μm, 1.6 μm C18, AUR2-25075C18A, Ion Opticks) with a flow rate of 0.35 μL/min for a total duration of 75 min (native-gel sample) or 43 min (SDS-PAGE sample) and ionized at 1.8 kV (native-gel sample) or 1.6 kV (SDS-PAGE sample) in the positive ion mode. The gradient was composed of 6% solvent B (3.5 min and 3 min for native-gel and SDS-PAGE samples), 6–25% B (42 min and 20 min for native-gel and SDS-PAGE samples), 25–40% B (14.5 min and 7 min for native-gel and SDS-PAGE samples), and 40–98% B (15 min and 13 min for native-gel and SDS-PAGE samples). The solvents used for native-gel samples were solvent A (2% ACN and 0.2% formic acid) and solvent B (80% ACN and 0.2% formic acid). The solvents used for SDS-PAGE gel samples were solvent A (0.1% formic acid in water) and solvent B (80% ACN and 0.1% formic acid). MS1 scans were acquired at the resolution of 60,000 from 375 to 1500 $m/z$, AGC target 3 × 10⁶, and maximum injection time of 15 ms for the native-gel sample, and 120,000 from 350 to 2000 $m/z$, AGC target 1 × 10⁶, and a maximum injection time of 50 ms for SDS-PAGE sample. For the native-gel sample, 12 abundant ions in MS2 scans were acquired at a resolution of 30,000, AGC target 1 × 10⁵, maximum injection time of 60 ms, and normalized collision energy of 28.

Dynamic exclusion was set to 30 sec, and ions with charges +1, +7, +8, and >+8 were excluded. The temperature of the ion transfer tube was 275 °C, and the S-lens RF level was set to 60. For the SDS-PAGE sample, MS2 scans were acquired in the ion trap using a fast scan rate on precursors with 2–7 charge states and quadrupole isolation mode (isolation window: 1.2 $m/z$) with a high-energy collisional dissociation (HCD, 30%) activation type. Dynamic exclusion was set to 30 s. The temperature of the ion transfer tube was 300 °C, and the S-lens RF level was set to 30.

## Analysis of MS proteomics data

MS2 fragmentation spectra were searched with Proteome Discoverer SEQUEST (version 2.5; Thermo Scientific, Waltham, MA) against in silico tryptic digested UniProt *Chlorobaculum tepidum* database. The maximum missed cleavages were set to 2. Dynamic modifications were set to oxidation on methionine (M, + 15.995 Da), protein N-terminal acetylation (+42.011 Da), and Met-loss (−130.040 Da). Carbamidomethylation on cysteine residues (C, + 57.021 Da) was set as a fixed modification. The maximum parental mass error was set to 10 ppm, and the MS2 mass tolerance was set to 0.6 Da. The false discovery threshold was set strictly to 0.01 using the Percolator Node validated by $q$-value. The relative abundance of parental peptides was calculated by integrating the area under the curve of the MS1 peaks using the Minora LFQ node. The results of the native-gel (overall complex) and SDS-PAGE (protein subunits with a size of less than 15 kDa) samples were listed in Supplementary Tables 1, 2.

## Negative-stain electron microscopy

Negatively stained specimens were prepared following the previously reported protocol[69]. A continuous carbon film-supported copper EM grid was glow-discharged for 15 sec using a Pelco easiGlow glow-discharge system (Ted Pella, Redding, CA). The protein sample was applied on the grid, air-dried, and stained with 0.75% (w/v) uranyl formate. Negatively stained specimens were imaged using a Philips CM12 or an FEI Tecnai TF20 transmission electron microscope (TEM) with a CCD camera. The image of negatively stained samples was used

to screen the protein quality for subsequent high-resolution structural study.

## Sample preparation for cryo-EM imaging and data collection

A holey-carbon C-flat grid (2/1; Protochips, Morrisville, NC) was glow-discharged for 15 seconds using a Pelco easiGlow glow-discharge system (Ted Pella, Redding, CA). 5 μL of 0.1 mg/mL protein sample was applied to the pretreated grid, and the excess solution was blotted away using a homemade plunge-freezer for 6 sec at room temperature in the ambient conditions. The grid was then quickly plunged frozen into liquid ethane and transferred to the grid storage. Particle homogeneity and ice thickness of the grid specimen were screened using an FEI Tecnai TF20 TEM (Thermo Fisher/FEI, Hillsborough, OR). Grids with thin ice and a homogeneous protein dispersion were used for subsequent cryo-EM data collection.

Electron movie data of the cryogenic specimens were collected using a Thermo Fisher/FEI Titan Krios TEM (Thermo Fisher/FEI, Hillsborough, OR) at an accelerating voltage of 300 keV with a Gatan K2 Summit direct electron detector (DED) camera (Gatan, Pleasanton, CA) in the Eyring Materials Center (EMC) at Arizona State University (ASU) (Tempe, AZ). The sizes of the C2 condenser and objective apertures were selected as 70 and 70 μm, respectively. Defocus range was set to $-0.8$ to $-2.5$ μm. Nominal magnification was set to ×47,259, resulting in a physical pixel size of 1.04 Å/pixel at the specimen level. The movie data were recorded at a counted rate of 8.47 e$^-$/pixel/second and a subframe rate of 200 msec in counting mode[70]. Total exposure was set to 6 seconds, accumulating to an electron dosage of 45.4 e$^-$/Å$^2$. Beam-image shift scheme was applied to accelerate data acquisition[71]. Data collection was automated using the customized SerialEM macros (version 3.9)[72]. Data without gain normalization was written in the LZW-compressed TIFF format and later unpacked on a computer workstation for image processing. The parameters used for data collection are listed in Supplementary Table 4.

## Image processing

A total of 32,898 electron movies were recorded. Image processing was generally conducted using the cryoSPARC software suite (version 3.3.1)[73]. Beam-induced motions were corrected using the 'Patch motion correction', and the image defocus and astigmatism was estimated using the 'Patch CTF estimation' function. The images were discarded if the computed contrast transfer function (CTF) did not fit the observed power spectrum beyond 10 Å$^{-1}$ resolution. 1,938,908 particles were automatically selected using Topaz (version 0.2.5)[74] and were curated using iterative two-dimensional (2D) classification procedures. Class members with a poorly aligned average were not selected for the subsequent image reconstruction. Selected particle images (1,753,711) were used to calculate ab initio three-dimensional (3D) reconstructions with $k = 5$ using stochastic gradient descent and branch-and-bound maximum-likelihood regularization. Three classes with discernible features of the reaction center were carried over for two rounds of heterogeneous refinement. The three 3D reconstructions of RC-FMO$_2$ (157,486), RC-FMO$_1$ (142,020), and RC (383,872) were generated and refined against their class members using homogeneous and non-uniform refinement[75]. Global and local CTF refinements were subsequently performed to improve the coefficients of higher-order aberrations, including per-particle defocus, trefoil, spherical aberration, and tetrafoil. The resolutions of the RC-FMO$_2$ and RC-FMO$_1$ reconstructions were 3.08 and 3.49 Å, respectively. The final resolution of the reconstruction was determined using gold-standard FSC (Fourier shell correlation) criteria at the cutoff of 0.143[76]. The $b$-factors used for the final map sharpening on the RC-FMO$_2$ and RC-FMO$_1$ densities were $-102.3$ and $-136.6$ Å$^2$, respectively. Directional anisotropy of the reconstruction was assessed using the 3DFSC method[77]. The local resolution of the reconstruction was estimated using a local windowed FSC method[78]. Further signal subtraction and focused local

refinement on the RC-FMO$_2$ reconstruction improved the resolutions to 2.92 Å (FMO1, PscA1, and PscB) and 3.06 Å (FMO2 and PscA2) for the local densities. The flowchart for single-particle image processing is illustrated in Supplementary Figs. 3, 4.

Density subtraction of the RC-FMO$_1$ density from the previous cryo-EM density (EMD-30069) was performed using the EMDA Python package[79]. The two maps were resampled, aligned, and normalized before subtraction. Subtracted values are presented in colors on the RC-FMO$_1$ density surface (Supplementary Fig. 6).

## Modeling

Previous atomic coordinates of the RC-FMO complex (PDB codes: 6M32) were used as a search template for the initial atomic model building[32]. Geometric configurations of the ligand molecules, such as BChl, Chl, iron-sulfur clusters (SF4), phospholipids (LMG and LHG), and carotenoids (F26 and F39), were optimized using AM1 (Austin Model 1) force field by the eLBOW program[80,81]. The template was first docked into RC-FMO$_2$, and RC-FMO$_1$ cryo-EM densities using the 'Fit in the Map' function in UCSF Chimera (version 1.16)[82]. The fit model was manually rebuilt using Coot (version 0.9.5)[83], and the coordinates of the PscC and the two accessory subunits, PscE and PscF (accession numbers: Q8KDI3 and G8KG87), were built in a de novo manner. The rebuilt models were then refined against the cryo-EM densities using the 'phenix.real_space_refine' program in the Phenix software package (version 1.20.1-4487)[84,85]. The refined models were validated using MolProbity[86]. The model refinement and validation statistics are listed in Supplementary Table 4. The electrostatic potential of the RC-FMO$_2$ surface was calculated using the Adaptive Poisson Boltzmann Solver (APBS) with the AMBER force field[87]. The atomic coordinate of the ferredoxin from *C. tepidum* (accession number: Q8KCZ6) was predicted and calculated using AlphaFold2 (version 2.0)[88]. The figures for the cryo-EM density maps and atomic models were prepared using UCSF Chimera or ChimeraX (version 1.2.5)[89].

## Calculating excitation energy transfer rates

Transfer rates between BChls were calculated according to Förster theory[47] using an R script. The framework for the spectral overlap integral and parameters for BChls was followed or obtained from previous methods[90,91]. The parameters for transition dipole, Stoke shift, and the full width at half maximum (FWHM) were set to 41 D$^2$ ($1.37 \times 10^{-28}$ C·m), 190 cm$^{-1}$, and 535 cm$^{-1}$, respectively.

## Reporting summary

Further information on research design is available in the Nature Research Reporting Summary linked to this article.

## Data availability

Cryo-EM density maps (MRC format) of the RC-FMO$_2$ and RC-FMO$_1$ protein complexes determined in this study were deposited in the Electron Microscopy Data Bank (EMDB) under accession numbers EMD-26471 (RC-FMO$_2$) and EMD-26469 (RC-FMO$_1$). Model coordinates were deposited in the Protein Data Bank (PDB) under accession numbers 7UEB (RC-FMO$_2$; https://doi.org/10.2210/pdb7ueb/pdb) and 7UEA (RC-FMO$_1$; https://doi.org/10.2210/pdb7uea/pdb). All the data are available in the EMDB and wwPDB databases or from the corresponding author upon request.

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

## Acknowledgements

We thank Chihiro Azai, Patricia Baker, and Hila Toporik for advising us on the procedure for culture growth. We very much appreciate Robert Blankenship and Kevin Redding for fruitful scientific discussion. We thank Dewight Williams and David Lowry for the EM assistance in the Eyring Materials Center (EMC) at Arizona State University (ASU). We acknowledge using the Titan Krios TEM at the EMC and the funding for the instrumentation by NSF MRI 1531991. A portion of this research was supported by NIH grant U24GM129547 and performed at the Pacific Northwest Center for Cryo-EM (PNCC) at Oregon Health & Science University (OHSU), Portland, OR, and accessed through EMSL (grid.436923.9), a DOE Office of Science User Facility sponsored by the Office of Biological and Environmental Research (PNCC proposal number: 160074). We thank Omar Davulcu for the EM assistance at the PNCC site. We acknowledge the funding support by Army Research Office (ARO) (W911NF2010321) to P.-L.C., the support by the DOE, Office of Basic Energy Sciences, Photosynthetic Systems Program (DE-FG02-07ER15902) to H.L., and the GPU device support by the NVIDIA GPU Grant Program to P.-L.C. We especially thank John C.H. Spence for initiating the idea for the quantum biology project, and we missed the days discussing sciences and making discoveries together.

## Author contributions

R.P., C.D.T., K.S., and Y.-P.P. performed cell culture and protein purification. R.P. and C.D.T. prepared frozen specimens for cryo-EM data collection. S.C. and K.-Y.C. performed the characterization of purified protein complexes. S.L., K.-W.C., T.-Y.W., H.L., and T.-F.C. performed mass spectrometric experiments and analyses. P.-L.C. processed cryo-EM image data and performed structural modeling, analyses, and interpretations. Y.M. performed the calculations to analyze possible excitation energy transfer pathways. R.P., Y.-P.P., and P.-L.C. prepared figures. B.N. and P.-L.C. initiated and supervised the project. R.P., Y.M., B.N., and P.-L.C. wrote the manuscript with input from all the authors.

## Competing interests

The authors declare no competing interests.
