## [Peer Review File · Nature Communications]

Molecular asymmetry of photosynthetic supercomplex from green sulfur bacteriaREVIEWER COMMENTS

Reviewer #1 (Remarks to the Author):

Please see the attached file.

The manuscript by Puskar et al. describes the cryo-EM structure of RC-FMO₂ which contains two FMO together with the cytochrome c subunit (PscC), two novel accessory protein subunits (PscE and PscF), and a linker pigment between FMO and the RC core that may mediate the energy transfer from FMO to RC. This is a significant advancement from the previous RC-FMO₁ structure that contains only one FMO trimeric complex and no PscC, PscE and PscF. The new subunits and pigments found in the structure are suggested to play roles in the energy transfer, electron transfer and/or assembly of the supercomplex, and the second FMO trimer is shown to have an orientation different from that of the first FMO trimer with no linker pigment, making it less efficient in energy transfer. This manuscript thus contains important information regarding the organization and function of the green algal RC-FMO supercomplex, and I recommend acceptance of the paper by the Journal.

Regarding the possible role of quinone, the authors did not find a density corresponding to the quinone molecule, in agreement with the previous studies. However, in the previous study (32), an unidentified density was found between A₀ and F_x, which is similar to the situation found in the heliobacterial RC (51). Does this density correspond to the PG molecule that the authors assigned near A₀? If yes, it would be good if the authors can comment on the position of this PG molecule (equivalent to the previously unidentified density or not?).

Some minor comments are listed below.

Lines 143-145: A loop of PscD (Q83-P108) was found to be shifted between the structures of RC-FMO₂ and the previous cryo-EM structure. Does this loop also shift between the RC-FMO₂ and RC-FMO₁ structure determined in this study?

Lines 161-162: “Mass spectrometry analysis characterized this subunit as a peptide enriched with lysine and arginine residues (Extended Data Table 1-2 and Fig. 1c).”. From which data can it be said that this subunit is enriched with lysine and arginine residues? There is no “Fig. 1c”.

Line 213: The word “Although” should be removed, and the two sentences should be linked by “and”.

Line 228: “A linker...” should be changed to “The linker...”.

Lines 263-264: “phosphatidyl glycerol” should appear in the first place it is abbreviated.

Supplementary Fig. 1: In the legend it was written “...shortened as GsbRC-FMO”. However, in the figure it reads “CtRC-FMO”. Please confirm. Also, it was written in the legend that “White contrast

are stained protein complexes and black is background.”. However, it seemed to be opposite in the panel c.

Reviewer #2 (Remarks to the Author):

Photosynthesis is a global process constantly present on Earth. The ultimate uniqueness of the processes lies in its ability to produce organic compounds from inorganic matter using light energy. Hence photosynthesis leads to primary organic production in the Biosphere.

The photosynthetic machinery of green sulfur bacteria (GSB) consists of a peripheral antenna chlorosome, light-harvesting Fenna-Matthews-Olson proteins (FMO), and a reaction centre (GsbRC). The GsbRC-FMO complex is a central photosynthetic complex of this bacteria involved and performing the photosynthetic reactions.

Working in accordance, GsbRC-FMO provides the energy transfer pathway from chlorosome via FMO trimeric protein complex to GsbRC. Previously only one trimeric FMO complex attached to the reaction centre of green sulfur bacteria *Chlorobaculum tepidum* was discovered and structurally characterised at 2.7 Å [PMID: 33214250].

A clear understanding of RC-FMO structure impacts the insight into the energy pathway and thus the photosynthetic reactions taking place in this green bacterium in total.

Here the research group (of Prof. Dr Chiu) is focused on the structural investigation of several types of RC-FMO complex from the same green sulphur bacteria – *Chlorobaculum tepidum*.

The current paper contains quite some novelties complementing the picture of the energy-transfer pathways in green bacteria - *C. tepidum*, namely:

- a) Identification of the cytochrome c subunit (PscC);
- b) Identification of PscE and PscF accessory protein subunits;
- c) Presence of novel second FMO complex in GsbRC-FMO, creating GsbRC-FMO2 – one RC core with two FMO trimeric complexes;
- d) Presence of new linker pigment between FMO1 and RC core.

The discovery of the new FMO complex became possible due to the mild solubilisation during the GsbRC-FMO2 isolation step - here, the researchers used the 0.85 mM DDM followed by size-exclusion chromatography and polyacrylamide gel electrophoresis vs 3% (w/v). In the previous research (PMID: 33214250), the triton X-100 as a detergent was used.

I have several questions as part of the review of this article.

1. Detergent.

Why did you choose DDM as a primary detergent for this experiment?

How did you decide on the detergent concentration: did you try several DDM concentrations during the isolation step?

How do you know that the isolated RC-FMO2 complex is stable and not artificially formed?

I would suggest doing the BN-PAGE with different DDM concentrations to observe the presence of the RC-FMO2 complex.

2. RC-FMOx

What additional evidence can you provide to show that RC-FMO2 is the largest possible complex?

Have you observed the variability within your RC-FMO2 3D classes (e.g., different connection angles of FMO trimers to PSII, etc.)?

How could you explain the lower resolution of the RC-FMO1 complex compared to the RC-FMO2 complex, which has almost the same particle number and applies no symmetry during the reconstruction?

Indeed, minor differences between the published RC-FMO1 structure and the one discussed in this article can be observed (position of BCL 808 in /a chain; BLC in /U chain etc.) See the attachment figure.

3. Grows conditions

What are the light conditions under which the *C. tepidum* was cultivated?

It is not clear from the Methods (550... Cells were grown anaerobically in glass bottles at 40°C under illumination).

Can the discovery of an additional FMO trimeric protein complex and the increase of carotenoid content be interpreted as a part of the adaptation quenching mechanism response to high-light conditions?

Could you clarify that in the text?

4. Presence of extra accessory proteins

Have you noticed smaller subunits in the cryo-EM movies? If yes – did you analyse them separately?

Do all of the reconstructions of the FC-FMO2 complex contain PscE and PscF?

Do all of the reconstructions of the FC-FMO1 complex missing PscE/PscC?

In the validation report, the FSC of FC-FMO2 (pg 154 - 372982_0_related_ms_6610589_rd4g5c.pdf) differs from the FC-FMO1 (pg 114 second part - 372982_0_related_ms_6610589_rd4g5c.pdf).

Such differences in FSC may be determined by structural differences/variabilities and differences in the masking step.

It seems that by using different masking, you may find the other densities in FC-FMO2, which can be possibly meaningful.

If we name the FMO1(old)-RC – the previously reported complex, and FMO1(old)-RC-FMO(new) – as the one you presented in this article, have you identified the RC-FMO(new) ONLY-complex within your dataset?

If not, that can give some hints in understanding the sequence formation of the RC-FMO2 complex. For that, the data need to be re-analysed closely. You can re-classify the data in relion/cryosparc using RC-FMO(new).

You can get the RC-FMO(new) by subtracting the FMO1(old) from the RC-FMO2 structure.

Some smaller corrections

168-170 – You say “..two low-resolution envelopes on the RC periplasmic surface can be identified in both our RC-FMO2 and RC-FMO1 densities (Extended Data Fig. 3)” – Please add the description and/or sign these densities on the correspondent figure. Or you can point to the other figure. The periplasmic and cytoplasmic sides are also not specified in this figure. Please check/modify correspondently.

219-221 – You say:” Different binding affinities between the two FMO trimers have been reported previously 32,37,40,49 , and combined with the structural observations, FMO1 seems to be more stably bound to the RC cytoplasmic surface than FMO2”.

Do you mean that the RC-FMO2 structure was already described before? Please clarify and check the references for this statement.

550 - Cells were grown anaerobically in glass bottles at 40°C under illumination – please ass the light growth conditions.

After main Fig. 5, all the figures are illustrated again. Please check and correct.

Extended Data Fig. 2 – “b” overlaps the 2D class averages – please change.

Extended Data Fig. 2 – Please add the Euler angle distribution of particles contributing to the final reconstructions.

Extended Data Fig. 6 – RC-FMO1 is specified as “grey”. Supposedly in fawn. Please check/modify.

Extended Data Fig. 4 b,c – Please add 90 rotation for both maps.

Extended Data Fig. 7 – please add the cyto/periplasmic side label to clarify the view on the figure.

Extended Data Fig. 11 – lipids colour is specified as “grey”. It does not seem to be evident from the figure. Please modify correspondently.

Overall, the reviewed article brings sufficient novelties to studying light energy transfer within green sulfur bacteria - *C. tepidum* by structural characterisation of novel RC-FMO2 photosynthetic protein complex. This article will interest the researcher from the photosynthetic protein community.

I recommend this article for the following publication in the NatureCommunicaion journal, after correspondent corrections.

Kind regards,

180°

180°

10 nm

-  EMD-30069
-  RC-FMO1
-  Difference

The colors represent the degree of similarity of the input map signal, being dark blue the smallest differences between input volumes and red the biggest differences as discribed PMID: 34469787

Reviewer #3 (Remarks to the Author):

Puskar et al. revealed the structure of photosynthetic supercomplex from green sulfur bacteria. The presented work improves our current knowledge about the organization and composition of the supercomplex, which was recently published by Chen et al. (2020) Science 370. Using a mild solubilization conditions, the authors were able to isolate a more complete assembly of the supercomplex. This enabled to identify positions of three other subunits PscC, E, F and a linker bacteriochlorophyll located between FMO trimer and PscA subunits, which have not been observed by Chen et al.

I have two comments that should be considered in the revised version.

1. They also observed asymmetric features in the structure, e.g. binding of only one PscE and PscF subunits or binding of just one bacteriochlorophyll a816 to RC dimer. In my opinion the authors should discuss more this asymmetry. There is still a possibility that the asymmetry (or at least a part of it) can be a result of isolation procedure and the situation in vivo can be different.

2. Discussion regarding the excitation energy transfer is based just on distance of the bacteriochlorophylls. But the transfer strongly depends also on the orientation factor of transition dipole moments of the two excited molecules. Calculation of FRET would provide more precise information about the excitation energy transfer pathways.

Other comments:

Line 110: Indicate in the text the identified subunits. It is not clearly described. Can you indicate in the Extended Data Table1 PscE and F subunits?

Lines 179: Do you expect two copies of PscE and PscF in vivo supercomplex?

Line 203: Is it possible the second bacteriochlorophyll was just lost during isolation?

Line 211: Where is PscD in the figure 3?

Figure 5: Please indicate FMO1 and FMO2 in the figure.

In summary, the work is of great interest to a broad scientific community, and I recommend accepting the manuscript for publication in Nature Communications after revision.

We very much appreciate the reviewers' insightful comments and suggestions. We list the point-by-point responses in the following along with the revised manuscript.

Reviewer #1 (Remarks to the Author):

Please see the attached file (attached in the following).

The manuscript by Puskar et al. describes the cryo-EM structure of RC-FMO₂ which contains two FMO together with the cytochrome c subunit (PscC), two novel accessory protein subunits (PscE and PscF), and a linker pigment between FMO and the RC core that may mediate the energy transfer from FMO to RC. This is a significant advancement from the previous RC-FMO₁ structure that contains only one FMO trimeric complex and no PscC, PscE and PscF. The new subunits and pigments found in the structure are suggested to play roles in the energy transfer, electron transfer and/or assembly of the supercomplex, and the second FMO trimer is shown to have an orientation different from that of the first FMO trimer with no linker pigment, making it less efficient in energy transfer. This manuscript thus contains important information regarding the organization and function of the green algal RC- FMO supercomplex, and I recommend acceptance of the paper by the Journal.

We appreciate Reviewer #1's interest in our work and his/her suggestions on the manuscript. We made essential edits to the revised manuscript and addressed Reviewer #1's questions below in this letter. We thank Reviewer #1 for raising the issues and giving us a chance to clarify some critical points that we did not explain clearly in the previous version.

Regarding the possible role of quinone, the authors did not find a density corresponding to the quinone molecule, in agreement with the previous studies. However, in the previous study (32), an unidentified density was found between A₀ and F_x, which is similar to the situation found in the heliobacterial RC (51). Does this density correspond to the PG molecule that the authors assigned near A₀? If yes, it would be good if the authors can comment on the position of this PG molecule (equivalent to the previously unidentified density or not?).

We thank Reviewer #1 for bringing this point up. We agree that we should include the structural inspection of our proposed PG density with those in the previous works. In our GsbRC-FMO₂ structure, the location of the phosphatidyl glycerol (PG) between A₀ and F_x corresponds to the unmodelled densities found in the previous GsbRC-FMO₁ cryo-EM structure (PDB code: 6M32; Chen *et al.*, 2020) (Extended Data Fig. 11a). The PG coordinate well fits this unmodelled density in the previous structure (Extended Data Fig. 11a).

In the heliobacterial RC (*HbRC*) structure (PDB code: 5V8K), the unmodelled density between A₀ and F_x is near the corresponding PG location of the GsbRC-FMO₂ (Gisriel *et al.*, 2017), but after superimposing the *HbRC* to our structure, our PG coordinate does not fit to this density (Extended Data Fig. 11b). The author later modeled their density as a PG lipid in their following paper (Gisriel *et al.*, 2021). Their PG shows a corresponding binding site but a different conformation than the one found in our GsbRC-FMO₂ complex.

We added the following text for our analysis in the revised manuscript (page 13, line 273):

“... within the *PscA* dimer (Fig. 4c). These PG lipids also fit the unmodelled densities between A_0 and F_x in the previous GsbRC-FMO₁ cryo-EM reconstruction³² (Extended Data Fig. 12a). In the HbRC electron density, two unmodelled densities were found in similar positions and were initially suggested to be isoprenyl phosphates⁵¹ and later modeled as PG lipids⁴⁵. However, our PG coordinates do not fit into the unmodelled densities in the HbRC structure (Extended Data Fig. 12b). They may present a slightly different conformation but possibly the same function as the PG lipids in the GsbRC-FMO₂ at this site. In our GsbRC-FMO₂ structure, ...”

Extended Data Fig. 12 | Unmodelled densities in the previous RC structures from *Chlorobaculum tepidum* and *Helio bacterium modesticaldum*. Structural superpositions of the RC-FMO₂ structure with **a**, the previous RC-FMO₁ structure from *C. tepidum* (PDB code: 6M32) and **b**, the homodimeric RC (HbRC) structure from *H. modesticaldum* (PDB code: 5V8K). RC-FMO₂, the previous RC-FMO₁, and HbRC are colored blue, green, and pink, respectively. The modeled PG lipid in the RC-FMO₂ structure is colored yellow. Contour levels for the RC-FMO₁ cryo-EM map and the HbRC 2Fo-Fc electron density map are 2.4 σ and 1.2 σ , respectively.

Chen, J.-H. *et al.* Architecture of the photosynthetic complex from a green sulfur bacterium. *Science* **370**, eabb6350 (2020).

Gisriel, C. *et al.* Structure of a symmetric photosynthetic reaction center-photosystem. *Science* **357**, 1021-1025 (2017).

Gisriel, C. *et al.* Recent advances in the structural diversity of reaction centers. *Photosyn Res* **149**, 329-343 (2021).

Some minor comments are listed below.

Lines 143-145: A loop of PscD (Q83-P108) was found to be shifted between the structures of RC- FMO₂ and the previous cryo-EM structure. Does this loop also shift between the RC-FMO₂ and RC- FMO₁ structure determined in this study?

We apologize for the confusion and revised Extended Data Fig. 6 as below. Both superpositions of RC-FMO₂ with RC-FMO₁ and the previous structure (PDB code: 6M32) show that the PscD loop (Q83-P108) shifts and has a conformational change (Extended Data Fig. 6e-f).

We revised Extended Data Fig. 6 and modified the manuscript as follows for clarification (page 7, line 137):

From “... When our RC-FMO₁ structure is superimposed with the previous cryo-EM structure, no significant structural differences could be identified (Extended Data Fig. 6a-c)³². However, our new RC-FMO₂ structure reveals for the first time the association of two FMO trimers, two PscC subunits, two newly discovered accessory subunits, and additional pigment molecules. Also, when the RC-FMO₂ structure is compared to the previous cryo-EM structure³², a loop of PscD (Q83-P108) exhibits a slight difference in conformation in the presence of the second FMO trimer, and this structural change potentially stabilizes the binding of the second FMO complex (Extended Data Fig. 6d).”

To

“... When our RC-FMO₁ structure is superimposed with the previous cryo-EM structure, no significant structural differences could be identified (Extended Data Fig. 6a-b)³². However, our new RC-FMO₂ structure reveals for the first time the association of two FMO trimers, two PscC subunits, two newly discovered accessory subunits, and additional pigment molecules (Fig. 1). Also, when the RC-FMO₂ structure is compared to the RC-FMO₁ or the previous cryo-EM structure³², a loop of PscD (Q83-P108) exhibits a slight conformational change in the presence of the second FMO trimer. This structural change could accommodate and stabilize the binding of the second FMO trimer (Extended Data Fig. 6c-f).”

Revised Extended Data Fig. 6 | Structural superpositions of the RC-FMO₂ and RC-FMO₁ assemblies. **a**, Density difference between the RC-FMO₁ and the previous RC-FMO₁ cryo-EM densities (EMD-30069). Subtracted values are shown in blue and red for positive and negative values, respectively, on the map surface. Superpositions of the structures: **b**, RC-FMO₁ and 6M32 (RMSD 1.060 Å); **c**, RC-FMO₂ and 6M32 (RMSD 1.123 Å); and **d**, RC-FMO₂ and RC-FMO₁ (RMSD 0.777 Å). RC-FMO₂, RC-FMO₁, and 6M32 are colored in light blue, fawn, and pink, respectively. **e** and **f**, Enlarged views of the

local regions squared in **c** (green) and **d** (dark red), respectively. FMO2 binding induces a slight loop conformational change on PscD (Q83-P108) (dashed orange square).

Lines 161-162: “Mass spectrometry analysis characterized this subunit as a peptide enriched with lysine and arginine residues (Extended Data Table 1-2 and Fig. 1c)”. From which data can it be said that this subunit is enriched with lysine and arginine residues? There is no “Fig. 1c”.

We apologize for the carelessness and confusion. We removed the redundant text of “~~and Fig. 1c~~” in the revised manuscript.

In the revised manuscript, to better demonstrate the abundance of the lysine and arginine residues of the found peptides, the PscE subunit, we added Extended Data Table 3 to show the percentages of individual residues and modified the manuscript as follows (page 8, line 159):

From “... *Mass spectrometry analysis characterized this subunit as a peptide enriched with lysine and arginine residues (Extended Data Table 1-2). ...*”

To “... *This membrane subunit contains a high proportion of charged residues (aspartate and glutamate:17%; lysine and arginine: 22%) (Extended Data Table 3). ...*”

Amino acid	Count	Percentage (%)
Alanine	4	6.8
Arginine	2	3.4
Asparagine	0	0.0
Aspartate	2	3.4
Cysteine	1	1.7
Glutamine	1	1.7
Glutamate	8	13.6
Glycine	3	5.1
Histidine	0	0.0
Isoleucine	3	5.1
Leucine	5	8.5
Lysine	11	18.6
Methionine	3	5.1
Phenylalanine	0	0.0
Proline	3	5.1
Serine	4	6.8
Threonine	3	5.1
Tryptophan	1	1.7
Tyrosine	1	1.7
Valine	4	6.8

Extended Data Table 3. Amino acid composition of the PscE subunit.

Line 213: The word “Although” should be removed, and the two sentences should be linked by “and”.

We thank for the suggestion. We changed the sentence as suggested as follows (page 11, line 225):
“... ~~Although~~ *The residues that contact FMO2 are more distant than those contacting FMO1, and the effective areas that contact FMO2 are much smaller than those for FMO1, ...*”

Line 228: “A linker...” should be changed to “The linker...”.

We changed the sentence as suggested as follows (page 11, line 240):
“The linker pigment was identified along the FMO1-PscA1 axis, but not the FMO2-PscA2 axis (Fig. 2a), ...”

Lines 263-264: “phosphatidyl glycerol” should appear in the first place it is abbreviated.

We appreciate the correction. We checked the manuscript and wrote the full name of “phosphatidyl glycerol” in the first place (page 13, line 273) and the abbreviated “PG” afterward.

Supplementary Fig. 1: In the legend it was written “...shortened as GsbRC-FMO”. However, in the figure it reads “CtRC-FMO”. Please confirm. Also, it was written in the legend that “White contrast are stained protein complexes and black is background.”. However, it seemed to be opposite in the panel c.

We apologize for the inconsistency and fixed the term as “GsbRC-FMO” in the revised figure. To better identify negatively stained particles in the image and resolve the confusion, we removed the original description, circled representative particles in orange, and added the following text in the figure legend of Extended Data Fig. 1:

“... ~~White contrast are stained protein complexes and black is background.~~ Representative particles for the target protein complexes are circled in orange. ...”

Revised Extended Data Fig. 1 | Characterization of membrane extraction from *Chlorobaculum tepidum*. **a**, Size-exclusion chromatographic (SEC) profile of the detergent-solubilized sample. **b**, SDS-PAGE of the eluted peak fraction from the SEC. The reaction center and FMO complexes of *C. tepidum* are shortened as GsbRC-FMO. Gel bands within the highlighted green rectangle (< 15 kDa) were excised and digested for further mass spectrometry analysis. **c**, Electron micrograph of the negatively stained protein sample. Representative particles for the target protein complexes are circled in orange. Scale bar indicates 60 nm.

Reviewer #2 (Remarks to the Author):

Photosynthesis is a global process constantly present on Earth. The ultimate uniqueness of the processes lies in its ability to produce organic compounds from inorganic matter using light energy. Hence photosynthesis leads to primary organic production in the Biosphere.

The photosynthetic machinery of green sulfur bacteria (GSB) consists of a peripheral antenna chlorosome, light-harvesting Fenna-Matthews-Olson proteins (FMO), and a reaction centre (GsbRC). The

GsbRC-FMO complex is a central photosynthetic complex of this bacteria involved and performing the photosynthetic reactions.

Working in accordance, GsbRC-FMO provides the energy transfer pathway from chlorosome via FMO trimeric protein complex to GsbRC. Previously only one trimeric FMO complex attached to the reaction centre of green sulfur bacteria *Chlorobaculum tepidum* was discovered and structurally characterised at 2.7 Å [PMID: 33214250].

A clear understanding of RC-FMO structure impacts the insight into the energy pathway and thus the photosynthetic reactions taking place in this green bacterium in total.

Here the research group (of Prof. Dr Chiu) is focused on the structural investigation of several types of RC-FMO complex from the same green sulphur bacteria – *Chlorobaculum tepidum*.

The current paper contains quite some novelties complementing the picture of the energy-transfer pathways in green bacteria - *C. tepidum*, namely:

- a) Identification of the cytochrome *c* subunit (PscC);
- b) Identification of PscE and PscF accessory protein subunits;
- c) Presence of novel second FMO complex in GsbRC-FMO, creating GsbRC-FMO2 – one RC core with two FMO trimeric complexes;
- d) Presence of new linker pigment between FMO1 and RC core.

The discovery of the new FMO complex became possible due to the mild solubilisation during the GsbRC-FMO2 isolation step - here, the researchers used the 0.85 mM DDM followed by size-exclusion chromatography and polyacrylamide gel electrophoresis vs 3% (w/v). In the previous research (PMID: 33214250), the triton X-100 as a detergent was used.

We very much appreciate Reviewer #2's comments and his/her viewpoints to improve our manuscript. We hope that the revised manuscript answers the questions and improves the unclarities of the previous version.

I have several questions as part of the review of this article.

1. Detergent.

Why did you choose DDM as a primary detergent for this experiment?

The previous cryo-EM structure of the GsbRC-FMO₁ showed a homodimeric RC core with only one FMO trimer attached and without cytochrome *c* subunits, PscC; however, the composition was not consistent with mass spectrometry and cross-linking results (Chen et al., 2020; He et al., 2014). Previous studies showed that the detergent used for membrane protein extraction may affect the stability of the membrane protein assembly (main references 1, 26, 35-39). Because the previous cryo-EM study had used Triton X-100 detergent during protein isolation and purification (Chen et al., 2020), we suspected whether Triton X-100 could be the reason for the lost protein subunits. Another non-ionic detergent, dodecyl maltoside (DDM), has also been reported to be efficient in isolating active membrane protein complex, including photosynthetic supercomplex, from *C. tepidum* (Aivaliotis et al., 2004; Feiler et al., 1992; Hager-Braun et al., 1995). DDM is also one of the most used detergents used for membrane protein extraction and solubilization in structural studies (40.6% of studies reported using DDM for membrane

protein extraction from the 2016 “Membrane Proteins of Known Structure” database) (Stetsenko and Guskov, 2017). Therefore, we chose DDM for the solubilization of our membrane protein complex.

Aivaliotis, M. *et al.* Isolation and characterization of an outer membrane protein of *Chlorobium tepidum*. *Photosynth Res* **79**, 161-166 (2004).

Chen, J.-H. *et al.* Architecture of the photosynthetic complex from a green sulfur bacterium. *Science* **370**, eabb6350 (2020).

Feiler, U. *et al.* Characterization of an improved reaction center preparation from the photosynthetic green sulfur bacterium *Chlorobium* containing the FeS centers FA and FB and a bound cytochrome subunit. *Biochemistry* **31**, 2608-2714 (1992).

Hager-Braun, C. *et al.* Stable photobleaching of P840 in *Chlorobium* reaction center preparations: Presence of the 42-kDa bacteriochlorophyll *a* protein and a 17-kDa polypeptide. *Biochemistry* **34**, 9617-9624 (1995).

He, G., Zhang, H., King, J.D., and Blankenship, R.E. Structural analysis of the homodimeric reaction center complex from the photosynthetic green sulfur bacterium *Chlorobaculum tepidum*. *Biochemistry* **53**, 4924-4930 (2014).

Stetsenko, A. and Guskov, A. An overview of the top ten detergents used for membrane protein crystallization. *Crystals* **7**, 197 (2017).

How did you decide on the detergent concentration: did you try several DDM concentrations during the isolation step?

We used the initial conditions learned from our previous experience with membrane protein purification. For each intermediate step, we quantified the detergent concentration by referring to the total protein amount needed for solubilization. We gradually decreased DDM concentrations from solubilization to the final gel-filtration step. The final DDM concentration was determined by testing the protein stability at the concentrations of its 0.5×, 1×, and 2× critical micelle concentration (CMC).

How do you know that the isolated RC-FMO₂ complex is stable and not artificially formed?

The RC-FMO₂ protein complex used in this study was directly isolated from *C. tepidum* inner membranes, which is in its native condition. It is unlikely that the RC-FMO₂ complex would randomly form after cell lysis. Also, many molecular projections of the RC-FMO₂ complex (~158k particles) were observed and measured, and their 2D projection densities were consistent and able to reconstruct a high-resolution 3D density map. Thus, it is less likely the isolated complex was artificially or randomly formed in the solution.

I would suggest doing the BN-PAGE with different DDM concentrations to observe the presence of the RC-FMO₂ complex.

We appreciate Reviewer #2's suggestion. To understand whether the DDM concentration affects RC-FMO₂ complex stability, we isolated the protein complex from *C. tepidum* membranes and mixed it with DDM detergent at the concentrations of 0.5× (0.085 mM), 1× (0.17 mM), 3× (0.51 mM), and 5× CMC (0.85 mM). We then performed the Blue-Native gel electrophoresis (BN-PAGE) analysis for these protein

samples (Fig. R1). The gel image did not show a clear band for indicated protein but a smeared contrast, and the results were reproducible for three times on different batches of purified samples (Fig. R1). This is probably because the detergent concentration affects the performance of the BN-PAGE (Reisinger and Eichacker, 2008). The resolution for the protein separation in the native gel may become poor when the DDM concentration is low. However, the smeared contrasts in different lanes showed that the proteins migrated to the bands at smaller molecular weights when DDM concentration increases (Fig. R1, lanes 3-5). With the highest DDM concentration ($5\times$ CMC), only lane 5 had the contrast in the position of the main band consistent with the molecular weight of RC-FMO₂. We did not attempt to use a higher DDM concentration for solubilization since higher DDM concentrations may affect the integrity of the RC-FMO₂ complex. The smeared bands in our gel were also shown in the previous report on the BN-PAGE analysis of the membrane proteins from *C. tepidum* (Alvaliotis et al., 2006).

On the other hand, during the solubilization test, we observed very little defined protein complexes at a DDM concentration of 0.085 mM ($0.5\times$ CMC, lane 2). This could be the reason why we did not detect any band in the native gel (Fig. R1, lane 2). Therefore, the BN-PAGE characterization could not give us a conclusive result of the complex at the lower range of DDM concentrations. Also, the Coomassie Brilliant Blue G-250 (CBB) used in Blue-native gel is weakly anionic, it could destabilize the binding of some peripheral protein subunits in a membrane protein complex (Schagger, 2001; Eubel et al., 2005).

Although the BN-PAGE result generated smeared bands for the purified protein complex, other biophysical characterizations were performed to show the identity of the protein complex at the final DDM concentration. Negative-stain EM showed the low-resolution blob features for the RC and two attached FMO trimers (Extended Data Fig. 1c). Mass spectrometry analysis of the native gel band of the purified proteins showed that the compositions of the supercomplex were PscA, PscB, PscC, PscD, PscE, PscF, and FMO (Extended Data Table 1). Also, previous biochemical characterization has shown the stoichiometry of two FMO trimers on the RC in DDM-solubilized protein complex (main reference 40). Therefore, the DDM concentrations we used for protein isolation are less likely to be harsh to the stability of the protein complex.

Fig. R1 | Blue-native polyacrylamide gel electrophoresis (BN-PAGE) analysis of purified GsbRC-FMO samples. Isolated protein samples were mixed with different DDM detergent concentrations. Gel bands were stained with Coomassie blue on a 3-12% Bis-Tris BN-PAGE gel. Lane 1: marker; lane 2-5: protein samples in DDM detergent concentrations of 0.085 mM (0.5× CMC), 0.17 mM (1× CMC), 0.51 mM (3× CMC), and 0.85 mM (5× CMC).

Alvaliotis, M., Karas, M., & Tsiotis, G. An alternative strategy for membrane proteome analysis of the green sulfur bacterium *Chlorobium tepidum* using Blue native PAGE and 2-D PAGE on purified membranes. *J. Proteome Res.* **6**, 1048-1058 (2007).

Eubel, H., Braun, H.-P., & Millar, A.H. Blue-native PAGE in plants: a tool in analysis of protein-protein interactions. *Plant Methods* **1**, 11 (2005).

Reisinger, V. and Eichacker, L.A. Solubilization of membrane protein complexes for blue native PAGE. *J. Proteomics* **71**, 277-283 (2008).

Schagger, H. Blue-native gels to isolate protein complexes from mitochondria. *Methods Cell Biol.* **65**, 231-244 (2001).

2. RC-FMO_x

What additional evidence can you provide to show that RC-FMO₂ is the largest possible complex?

We do not have sufficient evidence presented in this study to know whether the RC-FMO₂ complex is the largest protein complex in the membrane. However, the protein subunits identified in our RC-FMO₂ complex have fundamental components to drive light energy transfer and charge separation. Also, because of the limited RC cytoplasmic surface, it is not possible for the RC to accommodate an additional FMO trimer in the RC-FMO₂ supercomplex. Thus, two FMO trimers for one RC core are likely to be the maximum. Compared to the previous findings, the determined RC-FMO₂ complex in this study is currently the closest to a complete view of the GSB photosynthetic supercomplex.

Have you observed the variability within your RC-FMO₂ 3D classes (e.g., different connection angles of FMO trimers to PSII, etc.)?

We did not observe any structural variabilities between the 3D class averages of the RC-FMO₂ (Fig. R2). The inter-class variations are mostly the densities for the membrane portions. Only the membrane portion of Class 1 contains protein secondary structure features, and the other two are fragmented. It may be possible that the detergent-solubilized membrane portions are not structurally stable, generating structural heterogeneity that flattens the densities of the membrane portion or becomes fragmented.

Fig. R2 | Reconstructions of the 3D classification (heterogeneous refinement) on the RC-FMO₂ particle images.

How could you explain the lower resolution of the RC-FMO₁ complex compared to the RC-FMO₂ complex, which has almost the same particle number and applies no symmetry during the reconstruction?

One possible reason could be that the protein content of the RC-FMO₂ (two FMO trimers, PscE, and PscF) contributed to electron scattering is larger than that of the RC-FMO₁ (one FMO trimer). The molecular weight difference between the two complexes is about 135 kDa. The higher the protein scattering content will result in higher image alignment precision. Thus, within the same particle image dataset, the poses calculated by aligning the RC-FMO₂ particle images are more precise than those for RC-FMO₁. Thus, in the 3D reconstruction, the Fourier-shell correlation (FSC) of the RC-FMO₂ dataset shows the data consistency to a higher resolution than that of the RC-FMO₁ dataset.

Indeed, minor differences between the published RC-FMO₁ structure and the one discussed in this article can be observed (position of BCL 808 in /a chain; BLC in /U chain etc.) See the attachment figure.

We thank Reviewer #2 for the illustration that points out the density differences between the RC-FMO₁ and the previous structure. It gives us the idea to include this analysis in the revised manuscript. Although there are density variations around the PscA2-BChl-808 and FMO1-U-BChl, these structural changes may less likely affect the function of the supercomplex. We added a density subtraction panel in Extended Data Fig. 6 and modified its figure legend accordingly.

We also added the following text in the Method section to describe the processing of density subtraction (page 33, line 699):

“Density subtraction of the RC-FMO₁ density from the previous cryo-EM density (EMD-30069) was performed using the EMDA Python package (Warshamanage et al., 2022). The two maps were resampled, aligned, and normalized before subtraction. Subtracted values are presented in colors on the RC-FMO₁ density surface (Extended Data Fig. 6).”

Revised Extended Data Fig. 6 | Structural superpositions of the RC-FMO₂ and RC-FMO₁ assemblies. **a**, Density difference between the RC-FMO₁ and the previous RC-FMO₁ cryo-EM densities (EMD-30069). Subtracted values are shown in blue and red for positive and negative values, respectively, on the map surface. Superpositions of the structures: **b**, RC-FMO₁ and 6M32 (RMSD 1.060 Å); **c**, RC-FMO₂ and 6M32 (RMSD 1.123 Å); and **d**, RC-FMO₂ and RC-FMO₁ (RMSD 0.777 Å). RC-FMO₂, RC-FMO₁, and 6M32 are colored in light blue, fawn, and pink, respectively. **e** and **f**, Enlarged views of the

local regions squared in **c** (green) and **d** (dark red), respectively. FMO2 binding induces a slight loop conformational change on PscD (Q83-P108) (dashed orange square).

Warshamanage, R., Yamashita, K. & Murshudov, G.N. EMDA: A Python package for electron microscopy data analysis. *J. Struct. Biol.* **214**, 107826 (2022).

3. Grows conditions

What are the light conditions under which the *C. tepidum* was cultivated?

The cells *Chlorobaculum tepidum* were grown under a white-light illumination of 350 $\mu\text{mol photons/m}^2\cdot\text{sec}$ for two to four days. We modify our statement in the Method section as follows (page 28, line 572):

“... Cells were grown anaerobically in 1-L glass bottles at 40°C under white-light illumination of 350 $\mu\text{mol photons/m}^2\cdot\text{sec}$ for two to four days ...”

It is not clear from the Methods (550... Cells were grown anaerobically in glass bottles at 40°C under illumination).

We apologize for the confusion and modified the statement as follows (page 28, line 572):

“... Cells were grown anaerobically in 1-L glass bottles at 40°C under a white-light illumination of 350 $\mu\text{mol photons/m}^2\cdot\text{sec}$ for two to four days until the culture became dark green and turbid. Cells were harvested at OD_{750} of 2.8 cm^{-1} and were pelleted using centrifugation at 5,000 \times g for 7 minutes. ...”

Can the discovery of an additional FMO trimeric protein complex and the increase of carotenoid content be interpreted as a part of the adaptation quenching mechanism response to high-light conditions?

Could you clarify that in the text?

We appreciate Reviewer #2's questions, which could provide insight into how the supercomplex is adapted to a high-light condition. The carotenoids bound in the RC play a role in stabilizing the supercomplex structure or performing photoprotection in *C. tepidum* (main references 57-59). Therefore, the additional carotenoids we found in the RC-FMO₂ may play a role in energy dissipation to protect the cell from excessive light damage when the cells are grown in a high-light condition. However, we did not find additional carotenoids in the RC-FMO₁ complex.

Because the FMO trimer only contains bacteriochlorophyll *a* (BChl *a*) pigment, responsible for light energy transfer, it may be less likely that the assembly of the second FMO trimer is due to the adaptation to a high-light condition.

We added the following text in our revised manuscript (page 14, line 296):

“... Also, because our cells were grown in a high-light condition ($\sim 350 \mu\text{mol photons/m}^2/\text{sec}$), the additional carotenoids discovered in the RC-FMO₂ complex could assist in dissipating excessive light energy to adapt to the light stress.”

4. Presence of extra accessory proteins

Have you noticed smaller subunits in the cryo-EM movies? If yes – did you analyse them separately?

We did not find membrane protein subunits other than PscC, PscE, and PscF in our cryo-EM reconstructions. We have attempted to subtract the image intensities for PscE (6.49 kDa) or PscF (6.38 kDa) and focused refine their 3D densities. However, because their molecular masses are too small to contain sufficient scattering intensities, the results from the focused refinements were noisy and did not gain more high-resolution details.

Do all of the reconstructions of the FC-FMO₂ complex contain PscE and PscF?

The RC-FMO₂ reconstructions from different classes are shown in Fig. R2. They present various qualities and resolutions (“1”: 4.15 Å; “2”: 8.13 Å; and “3”: 9.10 Å). We can positively identify PscE and PscF from Class 1 reconstruction, but not from Class 2 or 3. The reconstructions from Class 2 or 3 do not have sufficient quality to recognize the protein features, partly due to poor image alignment on the intensities for the membrane portions.

Fig. R2 | Reconstructions of the 3D classification (heterogeneous refinement) on the RC-FMO₂ particle images.

Do all of the reconstructions of the FC-FMO₁ complex missing PscE/PscC?

The RC-FMO₁ reconstructions from different classes are shown in Fig. R3. As the same as the RC-FMO₂ data, the qualities of the reconstructions varied from class to class (“1”: 7.82 Å; “2”: 8.62 Å; “3”: 5.91 Å; and “4”: 8.37 Å), particularly for the membrane portions. The reconstruction from Class 3 shows the details of the secondary structures in the membrane portion, but we did not identify any densities for PscE or PscF. Reconstructions from Classes 1, 2, and 4 do not have sufficient quality to conclude whether PscE or PscF is present or not.

Fig. R3 | Reconstructions of the 3D classification (heterogeneous refinement) on the RC-FMO₁ particle images.

In the validation report, the FSC of FC-FMO2 (pg 154 - 372982_0_related_ms_6610589_rd4g5c.pdf) differs from the FC-FMO1 (pg 114 second part - 372982_0_related_ms_6610589_rd4g5c.pdf). Such differences in FSC may be determined by structural differences/variabilities and differences in the masking step.

It seems that by using different masking, you may find the other densities in FC-FMO2, which can be possibly meaningful.

We thank and appreciate Reviewer #2's comments. We agree that the masking procedure could neglect some densities for flexible or small protein subunits in the final reconstruction. We carefully checked the unmasked half maps of the RC-FMO₂ and RC-FMO₁ at different contour levels (Fig. R4; the densities were filtered to 3 Å⁻¹ resolution), but we could not find any additional densities that were not assigned.

Fig. R4 | Unmasked half maps of the RC-FMO₂ (purple) and RC-FMO₁ (yellow) complex at different contour levels. Maps were filtered to 3 Å⁻¹ resolution.

If we name the FMO1(old)-RC – the previously reported complex, and FMO1(old)-RC-FMO(new) – as the one you presented in this article, have you identified the RC-FMO(new) ONLY-complex within your dataset?

If not, that can give some hints in understanding the sequence formation of the RC-FMO2 complex. For that, the data need to be re-analysed closely. You can re-classify the data in rellion/cryosparc using RC-FMO(new).

You can get the RC-FMO(new) by subtracting the FMO1(old) from the RC-FMO2 structure.

We very much appreciate Reviewer #2's questions. Corresponding to the naming for the complexes in our report, the "FMO1(old)" and "FMO(new)" are the "FMO1" and "FMO2", respectively. We did not find

any densities of the RC-FMO2 (“RC-FMO(new)”) generated from the iterative 3D classification procedures. We only found the densities for the RC-FMO1 (“FMO1(old)-RC”) and the RC-FMO1-FMO2 (“FMO1(old)-RC-FMO(new)”).

We agree with Reviewer #2’s rationale that assembling FMO trimer complexes on the RC may be sequential (revised Fig. 5a), which could be indispensable to access this question in this work. Following Reviewer #2’s suggestion, we generate an RC-FMO2 density by subtracting the FMO1 density from that of the RC-FMO1-FMO2. The generated volume served as a reference with the RC-FMO1 and RC-FMO1-FMO2 for heterogeneous refinement without supplying a mask in the cryoSPARC program. The processing flowchart is illustrated in the following Extended Data Fig. 12 and added to the supplementary data. The group classified and refined by RC-FMO2 density occupies only 11.5% of the particle population, and its 3D class average did not generate high-resolution information or identifiable protein feature for the supercomplex. The volume without protein features may be generated by poorly aligned particle images. Thus, it is less likely that the FMO2 binds on the RC cytoplasmic surface alone without the presence of FMO1. Therefore, as Reviewer #2 suggests, we agree that the subunits are assembled into a supercomplex sequentially, but not randomly. It is very likely that the FMO1 binding is required for the FMO2 binding.

We propose a model for this sequential binding in revised Fig. 5 and add the following paragraph into the main text (page 16, line 338):

“Why the asymmetric arrangement of protein subunits in the GsbRC-FMO₂ supercomplex is established on a homodimeric RC core? Because our protein complex was directly extracted from the native membranes, the probability of obtaining a complex randomly formed by protein subunits is likely low. However, the protein isolation procedure may affect the stability of the protein assembly. Also, many projections of the protein complex imaged by cryo-EM could be used to calculate a high-resolution 3D reconstruction, which shows consistent observations across these molecular images. 3D image classification of our cryo-EM data results in the RC-FMO1-FMO2 (RC-FMO₂) and RC-FMO1 (RC-FMO₁) densities, but not RC-FMO2 (Extended Data Fig. 3). To test whether the RC-FMO2 could be possibly formed in the sample, we performed the supervised classification with the three densities and the result corroborates that the RC-FMO2 less likely presents in the population (Extended Data Fig. 14), implying that the FMO1 binding is required for FMO2 binding. Also, due to the different binding interfaces of the PscB with FMO1 and FMO2, it is unlikely that the two FMO trimers symmetrically bind to the RC. We thus propose a model of the supercomplex assembling process (Extended Data Fig. 15). The small membrane subunits, PscE and PscF, are transiently accessible to both sides of the homodimeric RC core. When a PscF binds to one side of the RC core, it may assist in stabilizing the FMO1 attachment, which subsequently determines the binding locations for PscE and FMO2 on the RC core (Extended Data Fig. 15). The densities for PscE or PscF on the other side of the RC core were not present in our cryo-EM density map. The reason could be that on the other side of the RC core, the assembly of these small subunits is not stable without interacting with a third FMO trimer, which is unlikely to be hosted in the supercomplex due to the limited RC cytoplasmic surface. Therefore, the overall asymmetric feature of the GsbRC-FMO₂ supercomplex is generated sequentially, but not randomly, through interactions between these proteins.”

Extended Data Fig. 14 | Supervised 3D classification or heterogeneous refinement using the densities of RC-FMO₁, RC-FMO₂, and RC-FMO₁-FMO₂. The density of RC-FMO₂ (light blue) was generated by the subtraction of FMO₁ density (grey) from the RC-FMO₁-FMO₂ density (trendy blue). The RC-FMO₁ density is shown in color purple.

Extended Data Fig. 15 | Proposed model for GsbRC-FMO₂ supercomplex assembly process. Color codes follow those used in Fig. 1. Arrows indicate the assembling sequence.

Some smaller corrections

168-170 – You say “..two low-resolution envelopes on the RC periplasmic surface can be identified in both our RC-FMO₂ and RC-FMO₁ densities (Extended Data Fig. 3)” – Please add the description and/or sign these densities on the correspondent figure. Or you can point to the other figure. The periplasmic and cytoplasmic sides are also not specified in this figure. Please check/modify correspondently.

We prepared an additional panel **b** and changed panel **b** to **c** in Extended Data Fig. 7 to show the densities for the two putative heme-binding domains in RC-FMO₂ and RC-FMO₁ (panel **b**). We also added the labels for the periplasmic and cytoplasmic sides in the figure (panel **c**).

We modify our manuscript accordingly in the following text (page 8, line 167):

“... can be identified in both our RC-FMO₂ and RC-FMO₁ densities (Extended Data Fig. 7b). ..., showing the position of the domain attachment to the PscA periplasmic surface close to the P840 center (Extended Data Fig. 7c). ...”

We also add the following text in the legend for Extended Data Fig. 7:

“...b, Low-contour cryo-EM densities for RC-FMO₂ (1.4σ) and RC-FMO₁ (1.8σ). Arrows indicate possible C-terminal heme-binding domains of PscC. c, ...”

Revised Extended Data Fig. 7 | Structure of the PscC subunit.

219-221 – You say:” Different binding affinities between the two FMO trimers have been reported previously 32,37,40,49 , and combined with the structural observations, FMO1 seems to be more stably bound to the RC cytoplasmic surface than FMO2”.

Do you mean that the RC-FMO2 structure was already described before? Please clarify and check the references for this statement.

We apologize for the confusion. We checked the references and modified our statement as follows (page 11, line 231):

“... Different binding affinities between the two FMO trimers have been reported or discussed previously^{32,37,40,49}. ~~and~~ Combined with our structural observation, FMO1 seems to be more stably bound to the RC cytoplasmic surface than FMO2.”

550 - Cells were grown anaerobically in glass bottles at 40°C under illumination – please ass the light growth conditions.

We added the illumination condition in the Method section as the following (page 28, line 572):

“... Cells were grown anaerobically in 1-L glass bottles at 40°C under white-light illumination of 350 $\mu\text{mol photons/m}^2\cdot\text{sec}$ for two to four days ...”

After main Fig. 5, all the figures are illustrated again. Please check and correct.

We apologize for the confusion, and we removed the duplication in the revised manuscript.

Extended Data Fig. 2 – “b” overlaps the 2D class averages – please change.

We improved the figure as follows:

Revised Extended Data Fig. 2 | Cryo-EM image analysis of membrane protein complexes from *C. tepidum*.

Extended Data Fig. 2 – Please add the Euler angle distribution of particles contributing to the final reconstructions.

In the revised manuscript, we added angular distribution plots for individual reconstructions in Extended Data Fig. 3.

Revised Extended Data Fig. 3 | Single-particle reconstruction of the GsbRC-FMO complex.

Extended Data Fig. 6 – RC-FMO1 is specified as “grey”. Supposedly in fawn. Please check/modify.

We fixed it by changing from “grey” to “fawn” as follows:

“... RC-FMO₂, RC-FMO₁, and 6M32 are colored in light blue, fawn, and pink, respectively. **d**, ...”

Extended Data Fig. 4 b,c – Please add 90° rotation for both maps.

We added additional panels to show the orthogonal view of both maps in Extended Data Fig. 4b-c.

Revised Extended Data Fig. 4 | Signal subtraction, local refinement, and local resolution estimation on the cryo-EM densities of the RC-FMO assemblies. b, RC-FMO₂ and c, RC-FMO₁ assemblies. Color bars are presented to indicate the local resolution values of the density maps.

Extended Data Fig. 7 – please add the cyto/periplasmic side label to clarify the view on the figure.

We added the labels for cytoplasmic/periplasmic sides of the membrane as follows (now in panel c in the revised Extended Data Fig. 7).

Revised Extended Data Fig. 7 | Structure of the PscC subunit.

Extended Data Fig. 11 – lipids colour is specified as “grey”. It does not seem to be evident from the figure. Please modify correspondently.

In the revised figure, we changed the color code for lipids from grey to crimson.

Revised Extended Data Fig. 13 | Membrane lipids play an important role in maintaining the integrity of the photosynthetic assembly.

Overall, the reviewed article brings sufficient novelties to studying light energy transfer within green sulfur bacteria - *C. tepidum* by structural characterisation of novel RC-FMO2 photosynthetic protein complex. This article will interest the researcher from the photosynthetic protein community.

I recommend this article for the following publication in the NatureCommunication journal, after correspondent corrections.

We very much thank for Reviewer #2's recommendation and suggestion. We hope you find our revised manuscript satisfactory.

Kind regards,

Reviewer #3 (Remarks to the Author):

Puskar et al. revealed the structure of photosynthetic supercomplex from green sulfur bacteria. The presented work improves our current knowledge about the organization and composition of the supercomplex, which was recently published by Chen et al. (2020) Science 370. Using a mild solubilization conditions, the authors were able to isolate a more complete assembly of the supercomplex. This enabled to identify positions of three other subunits PscC, E, F and a linker bacteriochlorophyll located between FMO trimer and PscA subunits, which have not been observed by Chen et al.

I have two comments that should be considered in the revised version.

We very much appreciate Reviewer #3's comments and modified the manuscript in the revised version. We also tried our best to answer the questions below and hope Reviewer #3 finds the responses satisfactory.

1. They also observed asymmetric features in the structure, e.g. binding of only one PscE and PscF subunits or binding or just one bacteriochlorophyll a816 to RC dimer. In my opinion the authors should discuss more this asymmetry. There is still a possibility that the asymmetry (or at least a part of it) can be a result of isolation procedure and the situation in vivo can be different.

We agree that the isolation procedure may affect the stability of the protein assembly. However, because the protein complexes were directly extracted from *C. tepidum* membranes, the protein subunits in the supercomplex could be disassembled but may less likely bind or re-bind to the supercomplex randomly by diffusion during the isolation process. Also, many recorded image projections of protein complexes could be used to calculate a high-resolution 3D reconstruction, which shows consistent observations on the compositions of the imaged protein complexes.

We also agree that the membrane portion, that is, the PscA-PscC-PscE-PscF complex, may be transiently symmetric since the RC core is homodimeric, providing redundant interfaces for protein-protein interactions. One possible reason for the asymmetric feature of the supercomplex is the limited space for FMO trimer binding, especially the FMO1. The FMO1 binding stabilizes the assembly of the PscF, PscB, and PscD and determines the binding locations of the PscE and FMO2 on the RC core. Another possibility could be that the PscB provides specific binding interfaces for FMO1 and FMO2, recruiting two FMO trimers asymmetrically (Fig. R5).

Fig. R5 | Binding of PscB with the two FMO trimer complexes. The shown two FMO trimer surfaces are the binding interfaces with the RC cytoplasmic surface. PscB (yellow) interacts extensively with FMO1 (dark green), but less engages with FMO2 (forest green).

We thank Reviewer #3's point of view and added the following paragraph in the revised manuscript (page 16, line 338):

“Why the asymmetric arrangement of protein subunits in the GsbRC-FMO₂ supercomplex is established on a homodimeric RC core? Because our protein complex was directly extracted from the native membranes, the probability of obtaining a complex randomly formed by protein subunits is likely low. However, the protein isolation procedure may affect the stability of the protein assembly. Also, many projections of the protein complex imaged by cryo-EM could be used to calculate a high-resolution 3D reconstruction, which shows consistent observations across these molecular images. 3D image classification of our cryo-EM data results in the RC-FMO1-FMO2 (RC-FMO₂) and RC-FMO1 (RC-FMO₁) densities, but not RC-FMO2 (Extended Data Fig. 3). To test whether the RC-FMO₂ could be possibly formed in the sample, we performed the supervised classification with the three densities and the result corroborates that the RC-FMO₂ less likely presents in the population (Extended Data Fig. 14), implying that the FMO1 binding is required for FMO2 binding. Also, due to the different binding interfaces of the PscB with FMO1 and FMO2, it is unlikely that the two FMO trimers symmetrically bind to the RC. We thus propose a model of the supercomplex assembling process (Extended Data Fig. 15). The small membrane subunits, PscE and PscF, are transiently accessible to both sides of the homodimeric RC core. When a PscF binds to one side of the RC core, it may assist in stabilizing the FMO1 attachment, which subsequently determines the binding locations for PscE and FMO2 on the RC core (Extended Data Fig. 15). The densities for PscE or PscF on the other side of the RC core were not present in our cryo-EM density map. The reason could be that on the other side of the RC core, the assembly of these small subunits is not stable without interacting with a third FMO trimer, which is unlikely to be hosted in the supercomplex due to the limited RC cytoplasmic surface. Therefore, the overall asymmetric feature of the GsbRC-FMO₂ supercomplex is generated sequentially, but not randomly, through interactions between these proteins.”

2. Discussion regarding the excitation energy transfer is based just on distance of the bacteriochlorophylls. But the transfer strongly depends also on the orientation factor of transition dipole moments of the two excited molecules. Calculation of FRET would provide more precise information about the excitation energy transfer pathways.

We agree that the calculation of the excitation energy transfer using FRET theory would provide insight into the transfer pathway within the supercomplex. We added Extended Data Fig. 9 in the revised manuscript.

We added a paragraph for analyzing the excitation energy transfer in our revised manuscript as follows (page 10, line 204):

“Calculating the transfer rates between FMO1 and RC using Förster theory further supported the significance of BChl-A816 in the transfer process. In addition to its close proximity, the orientation between the BChl-A816 and the terminal emitters of the FMO (subunit U) is highly favorable for energy transfer, leading to the fastest transfer rate in the entire FMO-RC interface (Extended Data Fig. 9 and Supplementary Data 1-2). A simple Förster treatment using uniform site energies yields the FMO-to-RC transfer rate of $\sim 0.13 \text{ ps}^{-1}$, significantly faster ($\sim 50\times$) than in vitro measurements (He et al., 2015), but in agreement with some recent in vivo estimations (Dostál et al., 2016; Ranjbar Choubeh et al., 2019). The discrepancy can be attributed to the lability of some pigments, such as BChl-A816, during isolation or to some limitations of the Förster treatment resulting in an overestimation of the transfer rate. Regardless of the absolute value of the transfer rate, the identification of BChl-A816 as a major contributor to energy transfer in this system should hold. It is also quite clear that FMO2 is not as well connected to the RC as FMO1, this is due to both the absence of BChl-A816 and slower overall transfer rates caused by the larger distance separating FMO2 and the RC (Extended Data Fig. 9 and Supplementary Data 1-2).”

We also added the detail of calculation in the Method section (page 34, line 722):

“Calculating excitation energy transfer rates. Transfer rates between BChls were calculated according to Förster theory (Sener et al., 2011) using an R script. The framework for the spectral overlap integral and parameters for BChls were followed or obtained from previous methods (Connolly et al., 1982; Gradinaru et al., 1998). The parameters for transition dipole, Stoke shift, and the full width at half maximum (FWHM) were set to $41 D^2$ ($1.37 \times 10^{-28} \text{ C}\cdot\text{m}$), 190 cm^{-1} , and 535 cm^{-1} , respectively.”

Extended Data Fig. 9 | Energy transfer rates between FMO and RC. **a**, and **b**, energy transfer rates (calculated using the Förster model) between all FMOs and RC BChls. Rates were calculated on BChl pairs closer than 40 Å (between FMO and RC) are shown as red lines. The radii of the connecting lines are scaled inversely to the transfer rates. Thicker lines represent faster rates. BChl-A816 alone accounts for more than 50% of the transfer between FMO1 and the RC (FMO1-BChl-A816 alone: 0.07 ps⁻¹; FMO1-RC: 0.11 ps⁻¹). It is also clear that the FMO1 is more efficiently connected to the RC than FMO2 (FMO2-RC: ~0.02 ps⁻¹). Transfer rates and orientation factors for individual BChl pairs are listed in Supplementary Data 1-2.

- Connolly, J.S., Janzen, A.F., & Samuel, E.B. Fluorescence lifetimes of chlorophyll *a*: solvent, concentration and oxygen dependence. *Photochem. Photobiol.* **36**, 559-563 (1982).
- Gradinaru, C.C. *et al.* The flow of excitation energy in LHCII monomers: implications for the structure model of the major plant antenna. *Biophys. J.* **75**, 3064-3077 (1998).
- He, G., Niedzwiedzki, D.M., Orf, G.S., Zhang, H., & Blankenship, R.E. Dynamics of energy and electron transfer in the FMO-reaction center core complex from the phototrophic green sulfur bacterium *Chlorobaculum tepidum*. *J. Phys. Chem. B* **119**, 8321-8329 (2015).
- Dostál, J., Pšenčík, J., & Zigmantas, D. *In situ* mapping of the energy flow through the entire photosynthetic apparatus. *Nat. Chem.* **8**, 705-710 (2016).
- Ranjbar Choubeh, R. *et al.* Efficiency of excitation energy trapping in the green photosynthetic bacterium *Chlorobaculum tepidum*. *Biochim. Biophys. Acta Bioenerg.* **1860**, 147-154 (2019).
- Sener, M. *et al.* Förster energy transfer theory as reflected in the structures of photosynthetic light-harvesting systems. *Chemphyschem* **12**, 518-531 (2011).

Other comments:

Line 110: Indicate in the text the identified subunits. It is not clearly described. Can you indicate in the Extended Data Table 1 PscE and F subunits?

We added the following texts for the identified subunits in the revised manuscript (page 6, line 110):

“... *Mass spectrometry (MS) analysis of the bands excised from blue native-gel electrophoresis (BNGE) indicated individual subunits that compose the photosynthetic complex, including two novel membrane subunits (Extended Data Table 1; Accession numbers: Q8KDI3 and Q8KG87). Negative-stain EM ...*”

Lines 179: Do you expect two copies of PscE and PscF *in vivo* supercomplex?

It could be possible that there are two copies of the PscE and PscF *in vivo* supercomplex and the binding of the second copy may be transient and not feasible for cryo-EM visualization. Also, if two copies of PscE and PscF bind to the supercomplex, the second copy may have less engagement with the FMO trimers, because the RC cytoplasmic surface is not large enough to host more than two FMO trimers. Without interacting with FMO complexes, the second copy of PscE and PscF may be less stable than the first one, leading to disassembly from the supercomplex. However, these will need to be verified by further characterization.

We added the following paragraph to clarify this point (page 16, line 338):

“Why the asymmetric arrangement of protein subunits in the GsbRC-FMO₂ supercomplex is established on a homodimeric RC core? Because our protein complex was directly extracted from the native membranes, the probability of obtaining a complex randomly formed by protein subunits is likely low. However, the protein isolation procedure may affect the stability of the protein assembly. Also, many projections of the protein complex imaged by cryo-EM could be used to calculate a high-resolution 3D reconstruction, which shows consistent observations across these molecular images. 3D image classification of our cryo-EM data results in the RC-FMO1-FMO2 (RC-FMO₂) and RC-FMO1 (RC-FMO₁) densities, but not RC-FMO2 (Extended Data Fig. 3). To test whether the RC-FMO2 could be possibly formed in the sample, we performed the supervised classification with the three densities and the result corroborates that the RC-FMO2 less likely presents in the population (Extended Data Fig. 13), implying that the FMO1 binding is required for FMO2 binding. Also, due to the different binding interfaces of the PscB with FMO1 and FMO2, it is unlikely that the two FMO trimers symmetrically bind to the RC. We thus propose a model of the supercomplex assembling process (Fig. 5a). The small membrane subunits, PscE and PscF, are transiently accessible to both sides of the homodimeric RC core. When a PscF binds to one side of the RC core, it may assist in stabilizing the FMO1 attachment, which subsequently determines the binding locations for PscE and FMO2 on the RC core (Fig. 5a). The densities for PscE or PscF on the other side of the RC core were not present in our cryo-EM density map. The reason could be that on the other side of the RC core, the assembly of these small subunits is not stable without interacting with a third FMO trimer, which is unlikely to be hosted in the supercomplex due to the limited RC cytoplasmic surface. Therefore, the overall asymmetric feature of the GsbRC-FMO₂ supercomplex is generated sequentially, but not randomly, through interactions between these proteins.”

Line 203: Is it possible the second bacteriochlorophyll was just lost during isolation?

We carefully inspected the densities within protein subunits and at protein-protein interfaces around PscA2, PscC, PscF, and FMO2. The spaces between protein subunits are mostly filled with lipid molecules, leaving no sufficient space for additional BChls to bind. We agree that protein isolation may influence the binding of bacteriochlorophyll to protein, but in this case, it is less likely that along the PscA2-FMO2 axis, the correspondingly second BChl exists at protein interfaces.

To clarify this, we modified our text (page 10, line 201):

From “... *This linker BChl is not found at the FMO2 and PscA2 interface and thus ...*”

To

“... *This linker BChl is not found at the FMO2 and PscA2 interface, and the spaces between PscA2, PscC, PscE, and FMO2 are filled with lipid molecules, leaving no space for additional BChls. Thus, ...*”

Line 211: Where is PscD in the figure 3?

We added the labels “PscD” in the revised Fig. 3 (main and panel d).

Revised Fig. 3 | Asymmetric binding of the two FMO complexes on the RC cytoplasmic surface.

Figure 5: Please indicate FMO1 and FMO2 in the figure.

We added the labels, “FMO1” and “FMO2”, in the revised Fig. 5.

Revised Fig. 5 | Proposed energy transfer and ETC pathways of the GsbRC-FMO₂ photosynthetic supercomplex. Model for the energy transfer in the GsbRC-FMO₂ supercomplex. Yellow arrows are possible light excitation energy transfer pathways. Blue arrows indicate the direction of the electron transport along the chlorophylls and iron-sulfur clusters.

In summary, the work is of great interest to a broad scientific community, and I recommend accepting the manuscript for publication in Nature Communications after revision.

REVIEWERS' COMMENTS

Reviewer #1 (Remarks to the Author):

The authors have addressed my concerns adequately, and I have no further questions.

Reviewer #2 (Remarks to the Author):

The group of Dr Chiu has done tedious revision work, performing the additional analysis and changing the manuscript correspondently. The provided answers satisfied most of my concerns.

I believe the manuscript is now ready for publication in Nature Communications journal.

Kind regards,

Dmitry Semchonok

Reviewer #3 (Remarks to the Author):

In the revised version of the manuscript, the authors have sufficiently addressed my comments. I agree with the proposed changes and responses.